# Sound Heuristic Search Value Iteration for Undiscounted POMDPs with Reachability Objectives

**Qi Heng Ho**[1]    **Martin S. Feather**[2]    **Federico Rossi**[2]    **Zachary Sunberg**[1]    **Morteza Lahijanian**[1]

[1]Department of Aerospace Engineering Sciences, University of Colorado Boulder, Boulder, Colorado, USA
[2]Jet Propulsion Laboratory, California Institute of Technology, Pasadena, California, USA

## Abstract

Partially Observable Markov Decision Processes (POMDPs) are powerful models for sequential decision making under transition and observation uncertainties. This paper studies the challenging yet important problem in POMDPs known as the (indefinite-horizon) Maximal Reachability Probability Problem (MRPP), where the goal is to maximize the probability of reaching some target states. This is also a core problem in model checking with logical specifications and is naturally undiscounted (discount factor is one). Inspired by the success of point-based methods developed for discounted problems, we study their extensions to MRPP. Specifically, we focus on trial-based heuristic search value iteration techniques and present a novel algorithm that leverages the strengths of these techniques for efficient exploration of the belief space (informed search via value bounds) while addressing their drawbacks in handling loops for indefinite-horizon problems. The algorithm produces policies with two-sided bounds on optimal reachability probabilities. We prove convergence to an optimal policy from below under certain conditions. Experimental evaluations on a suite of benchmarks show that our algorithm outperforms existing methods in almost all cases in both probability guarantees and computation time.

## 1 INTRODUCTION

Partially Observable Markov Decision Processes (POMDPs) are powerful models for sequential decision making in which the agent has both transition uncertainty and partial observability Sondik [1978], Kaelbling et al. [1998]. The goal in a POMDP planning problem is to compute a policy that optimizes for some objective, typically expressed as a reward function or logical specification, e.g., linear temporal logic (LTL) or probabilistic computation tree logic Baier and Katoen [2008]. POMDP problems are notoriously hard (due to the curse of dimensionality and history), and finding an optimal policy for them is undecidable [Madani et al., 2003]. Hence, to enable tractability, simple objectives are usually considered by either fixing a finite-time horizon or posing discounting. For such objectives, effective techniques have been developed, which can provide approximate solutions fast [Shani et al., 2013]. However, a crucial problem in planning under uncertainty and probabilistic model checking is to find a policy that maximizes the probability of reaching a set of target states without knowing *a priori* how many steps it may take. This is known as the (indefinite-horizon) Maximal Reachability Probability Problem (MRPP) [De Alfaro, 1998]. This paper focuses on MRPP for POMDPs and aims to develop an efficient algorithm with optimality guarantees for MRPP.

For infinite horizon discounted problems, point based methods [Pineau et al., 2003, Smith and Simmons, 2005, Kurniawati et al., 2009] approximate the value function by incrementally exploring the space of reachable beliefs. Trial-based belief exploration algorithms such as Heuristic Search Value Iteration (HSVI2) [Smith and Simmons, 2005] and its extensions have been shown to be the most effective. These methods utilize two-sided bounds to heuristically search for a near-optimal policy via tree search, and can efficiently solve moderately large POMDPs in both finite and discounted infinite-horizon settings to arbitrary precision. However, these techniques have not been studied to address (undiscounted indefinite-horizon) MRPP. Recently, approaches have been proposed to provide under-approximations on reachability probabilities for MRPP [Bork et al., 2022, Andriushchenko et al., 2022, 2023]. Although the under-approximations are shown to be tight empirically, there is no way to ascertain how close the approximations are or whether the computed policy has converged to optimality.

Taking inspiration from the success of trial-based belief

exploration for discounted POMDPs and the goal of designing an algorithm that can efficiently attain tight two-sided bound approximations, this paper studies the effectiveness of trial-based belief exploration for POMDPs with MRPP objective. To this end, we analyze the drawbacks of discounted POMDP trial-based search when applied to MRPP, and propose an algorithm that leverages the strength of trial-based belief exploration while addressing these drawbacks.

Our proposed algorithm is a trial-based belief exploration approach that maintains and improves two-sided bounds on the maximal probability of satisfying a reachability objective. Its use of forward exploration using these bounds allows informed exploration of the relevant areas of the belief space to improve search efficiency. Improving bounds at one belief can also improve bounds at other beliefs. We propose new heuristics for trial-based search tailored to MRPP, and discuss techniques to ensure improvability of both bounds during search. We prove the asymptotic convergence of the policy from below under some conditions. Experimental evaluations show the applicability of our approach to compute tight lower and upper bounds simultaneously that improve over time, converging to the optimal solution for several moderately sized POMDPs. Results show that trial-based exploration allows for efficient search, outperforming state-of-the-art belief-based approaches. Further, our approach is highly competitive, obtaining two-sided bounds that can be tighter than that of existing solutions which generally compute either a lower *or* upper bound, not both.

In short, the contributions of the paper are: (i) an analysis of theoretical and practical issues when applying discounted-sum algorithms to MRPP, (ii) an efficient algorithm that simultaneously computes sound upper and lower bounds for maximal reachability probabilities of POMDPs in an anytime manner, (iii) proof of asymptotic convergence of the lower bound to the optimal reachability probability value, and (iv) a suite of benchmark comparisons that show our algorithm outperforms existing methods in almost every case both in tightness of the bound and computation time.

**Related Work**    Algorithms to solve POMDPs with infinite horizon discounted properties have been extensively studied in the literature [Lauri et al., 2023, Shani et al., 2013]. A major bottleneck in those methods is the curse of dimensionality and history. To alleviate it, point-based methods such as Perseus [Spaan and Vlassis, 2005], HSVI2 [Smith and Simmons, 2005], SARSOP[Kurniawati et al., 2009], and PLEASE [Zhang et al., 2015] approximate the value function by incrementally exploring the space of reachable beliefs. These algorithms have been shown to be effective for moderately large discounted-sum POMDPs. They have been applied to the MRPP and POMDPs with LTL specifications [Bouton et al., 2020, Kalagarla et al., 2022, Yu et al., 2024], but their theoretical properties only hold for discounted versions of such problems. In this work, we study the drawbacks of point based methods for MRPP,

and propose an algorithm based on them to overcome these drawbacks and provide theoretical soundness.

It has been shown that MRPP is a special type of Stochastic Shortest Path Problem (SSP) with a specific non-negative reward structure [De Alfaro, 1998]. Horák et al. [2018] introduces a similar problem called Goal-POMDP, which is is an SSP with only positive costs and a set of goal states. The objective of Goal-POMDP is to minimize the expected total cost until the goal set is reached. Similar to this work, Horák et al. [2018] proposes extensions of HSVI2 to solve Goal-POMDPs. However, Goal-POMDP is different from MRPP because the assumption of positive cost and that the goal state is reachable from every state cannot be applied to MRPP. Hence, algorithms for Goal-POMDPs cannot directly be used to solve MRPP.

The works closest to ours are belief-based approaches [Norman et al., 2017, Bork et al., 2022, 2020]. To the best of our knowledge, the only method which computes two-sided bounds with convergence guarantees is PRISM Norman et al. [2017]. However, the approach is not scalable for larger POMDP problems, computing loose bounds in practice. Bork et al. [2022] computes under-approximations by expanding beliefs in a breadth-first search manner, and adding them to a constructed belief MDP according to some heuristic. Beliefs not added are *cut-off*, and values from a pre-computed policy are used from cut-off beliefs. In a similar manner, Bork et al. [2020] compute over-approximations using breadth-first belief exploration and cut-offs. However, their technique relies heavily on good pre-computed policies, and requires searching a large part of the belief space to obtain good policies. On the contrary, our algorithm performs heuristic trials in a depth-first manner using two-sided bounds, directing the search more efficiently.

An orthogonal approach to MRPP on POMDPs is to directly compute policies as Finite State Controllers (FSCs) [Andriushchenko et al., 2022]. Andriushchenko et al. [2022] uses inductive synthesis to search for FSCs. They can find good small-memory policies relatively quickly, but they suffer when they require memory. Andriushchenko et al. [2023] proposes an approach which integrates inductive synthesis with belief-based approaches to extract the strengths of both techniques. Generally, these methods compute under-approximations in an anytime manner, and cannot detect when or if a near-optimal policy has been found. Our method, on the other hand, computes both lower and upper bounds, providing sub-optimality bounds and means for near-optimal policy guarantees.

# 2   PRELIMINARIES AND PROBLEM FORMULATION

**Partially Observable Markov Decision Processes**    We focus on POMDPs with the following definition.

**Definition 1** (POMDP). *A Partially Observable Markov Decision Process (POMDP) is a tuple $\mathcal{M} = (S, A, O, T, Z, b_0)$, where: $S, A,$ and $O$ are finite sets of states, actions, and observations, respectively, $T : S \times A \times S \rightarrow [0,1]$ is the transition probability function, $Z : S \times A \times O \rightarrow [0,1]$ is the probabilistic observation function, and $b_0 \in \Delta(S)$ is an initial belief, where $\Delta(S)$ is the probability simplex (the set of all probability distributions) over $S$.*

We denote the probability distribution over states in $S$ at time $t$ by $b_t$ and the probability of being in state $s \in S$ at time $t$ by $b_t(s)$.

The evolution of an agent according to a POMDP model is as follows. At each time step $t \in \mathbb{N}_0$, the agent has a belief $b_t$ of its state $s_t$ as a probability distribution over $S$ and takes action $a_t \in A$. Its state evolves from $s_t \in S$ to $s_{t+1} \in S$ according to probability $T(s_t, a_t, s_{t+1})$, and it receives an observation $o_t \in O$ according to observation probability $Z(s_{t+1}, a_t, o_t)$. The agent then updates its belief recursively. That is for $s_{t+1} = s'$,

$$b_{t+1}(s') \propto Z(s', a_t, o_t) \sum_{s \in S} T(s, a_t, s') b_t(s). \quad (1)$$

Then, the process repeats.

The agent chooses actions according to a *policy* $\pi : \Delta(S) \rightarrow A$, which maps a belief $b$ to an action. Typically, the agent is given a reward function $R : S \times A \rightarrow \mathbb{R}$, which is the immediate reward of taking action $a_t$ at state $s_t$ and transitioning to $s_{t+1}$. A POMDP can be reduced to an MDP with an infinite number of states, whose states are the beliefs $B = \{b \in \Delta(S)\}$. This MDP is called a *belief MDP* [Åström, 1965]. Let $R(b, a) = \mathbb{E}[R(s, a)]$. When given a discount factor $\gamma \in [0, 1]$, the *expected discounted-sum of rewards* that the agent receives under policy $\pi$ starting from belief $b_t$ is

$$V^\pi(b_t) = \mathbb{E}\Big[ \sum_{j=t}^{\infty} \gamma^{j-t} R(b_j, \pi(b_j)) \mid b_t, \pi \Big]. \quad (2)$$

The objective of discounted POMDP planning is typically to find a policy that maximizes $V^\pi(b_0)$ to some threshold $\epsilon$.

The optimal value function $V^*$ for a POMDP can be under-approximated arbitrarily well by a piecewise linear and convex function [Sondik, 1978], $V^*(b) \geq \max_{\alpha \in \Gamma^*}(\alpha^T b)$, where $\Gamma^*$ is a finite set of $|S|$-dimensional hyperplanes, called $\alpha$-vectors, representing the optimal value function.

**Trial-based Value Iteration for Discounted POMDPs**

There exists mature literature on algorithms for discounted-sum POMDP problems for discount factor $\gamma < 1$. Among discounted-sum POMDP algorithms that provide finite time convergence guarantees, trial-based heuristic tree search

[Smith and Simmons, 2005, Kurniawati et al., 2009, Zhang et al., 2015] generally exhibit the best performance. These algorithms typically maintain and refine upper and lower bounds on the value functions, and they explore the reachable belief space through repeated trials over a constructed belief tree.

The basic ingredients of these algorithms are policy representation, action selection, observation/belief selection, backup function, and trial termination criteria. We briefly describe HSVI2 [Smith and Simmons, 2005], a trial-based heuristic search algorithm with these basic ingredients.

HSVI2 maintains a set of upper $V^U$ and lower $V^L$ bounds on the optimal value function. Lower bounds are represented as $\alpha$-vectors and upper bounds are computed using an upper bound point set. Trials are conducted in a depth-first manner. At each belief $b_t$, the action with the highest $Q$ upper bound is chosen for expansion using the IE-MAX heuristic:

$$a^* = \arg\max_a \{R(b, a) + \mathbb{E}[V^U(b_{t+1})]\}. \quad (3)$$

Then, an observation is selected by computing the successor belief $b_{t+1}$ with the highest weighted excess uncertainty (WEU):

$$\text{WEU}(b_t, t, \epsilon) = [V^U(b_t) - V^L(b_t) - \epsilon\gamma^{-t}], \quad (4)$$
$$o^* = \arg\max_o [P(o|b, a) \cdot \text{WEU}(b_{t+1}, t+1, \epsilon)]. \quad (5)$$

HSVI2 terminates a trial when $V^U(b_t) - V^L(b_t) \leq \epsilon\gamma^{-t}$. After each trial, a Bellman backup is performed over the belief states sampled, improving the lower and upper bound sets. When a discounted ($0 \leq \gamma < 1$) POMDP is given, HSVI2 provably converges to an $\epsilon$-optimal approximation of $V^*(b_0)$ with sound two-sided bounds.

**Problem Definition** The problem we are interested in is the undiscounted indefinite-horizon Maximal Reachability Probability Problem (MRPP). The agent is given a POMDP $\mathcal{M}$ with a set of target states $\text{T} \subseteq S$. We define the *reachability probability* $P^\pi_{\mathcal{M}}(\Diamond\text{T})$

to be the probability of reaching $\text{T}$ under policy $\pi$ from an initial belief $b_0$ for the POMDP $\mathcal{M}$. An *optimal policy* that maximizes the reachability probability is: $\pi^* = \arg\sup_\pi P^\pi_{\mathcal{M}}(\Diamond\text{T})$.

**Problem 1** ($\epsilon$-MRPP). *Given a POMDP $\mathcal{M}$, a set of target states $\text{T} \subseteq S$, and a regret bound $\epsilon \in (0, 1]$, find policy $\hat{\pi}$ that is $\epsilon$-optimal, i.e.,*

$$P^{\pi^*}_{\mathcal{M}}(\Diamond\text{T}) - P^{\hat{\pi}}_{\mathcal{M}}(\Diamond\text{T}) \leq \epsilon. \quad (6)$$

Probabilistic reachability values can be computed by augmenting the POMDP with an absorbing state $S \cup \{s_\text{T}\}$ and action $A \cup \{a_\text{T}\}$ such that transition probabilities $T(s, a, s') = 1$ if $s \in \text{T} \cup \{s_\text{T}\}$, $a = a_\text{T}$, and $s' = s_\text{T}$;

otherwise $T(s, a, s') = 0$. Then, by defining a reward function that assigns a reward of 1 to the augmented transitions to $s_T$ and otherwise 0 (i.e., $R_{\rm rp}(s, a) = 1$ if $s \in T$ and $a = a_T$, otherwise $R_{\rm rp}(s, a) = 0$) the undiscounted ($\gamma = 1$) expected cumulative reward of a policy is equivalent to its reachability probability [De Alfaro, 1998], i.e., for $\gamma = 1$,

$$P_{\mathcal{M}}^{\pi}(\Diamond T) = V^{\pi}(b_0) = \mathbb{E}\Big[\sum_{t=0}^{\infty} \gamma^t R_{\rm rp}(b_t, \pi(b_t)) \mid b_0, \pi\Big]. \tag{7}$$

**Remark 1.** *In many cases, one may want to answer the question of whether there exists a policy that has a reachability probability that exceeds a given threshold. Solutions to Problem 1 also allows one to answer such questions.*

**Remark 2.** *An algorithm for MRPP can also be used for problems in which the agent is tasked with maximizing the probability of satisfying temporal logic specifications, such as syntactically co-safe Linear Temporal Logic (cs-LTL) [Kupferman and Vardi, 2001] or LTL over finite traces (LTLf) [De Giacomo and Vardi, 2013]. These objectives can be converted into MRPP by planning in the product space of the POMDP and a Deterministic Finite Automaton (DFA) representation of the temporal logic formula.*

# 3  FROM DISCOUNTED-SUM TO REACHABILITY PROBABILITIES

Given the success of trial based tree search algorithms in discounted POMDPs and their application to probabilistic reachability problems [Bouton et al., 2020], a natural question that arises is whether these algorithms can directly approximate MRPP well. As it turns out, they unfortunately lose their desired theoretical properties, and there are problems in which these algorithms perform poorly. We discuss the issues that arise when applying these algorithms to MRPP. We focus on HSVI2 [Smith and Simmons, 2005], but the arguments presented also hold for other trial-based discounted-sum POMDP algorithms, e.g., [Kurniawati et al., 2009, Zhang et al., 2015].

**Incorrect converged solution if $\gamma < 1$ is used.** Discounting causes trial-based search to converge to an under-approximation of the optimal probability. This is a strict under-approximation for all but trivial problems. More importantly, the admitted upper bound is incorrect when $\gamma < 1$.

**Proposition 1.** *The optimal value $V^{\pi^{\gamma}}(b_0)$ computed via Eq. (7) with discount $\gamma < 1$ strictly under-approximates the optimal probability value $P_{\mathcal{M}}^{\pi^*}(\Diamond T)$ for Problem 1 if it takes more than one time step to reach $T$ in the POMDP.*

*Proof.* Let $\pi^*$ and $\pi^{\gamma}$ be the policies that maximize Eq. (7)

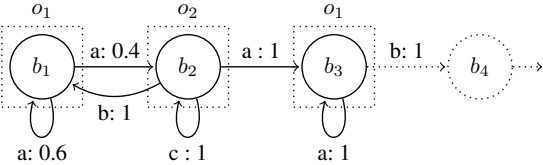

Figure 1: Belief MDP with loops that HSVI2 is ineffective on. The lower and upper bound values are initially $[0, 1]$ for all belief states. $b_4$ is initially not yet explored.

with $\gamma = 1$ and $\gamma < 1$, respectively. Then,

$$V^{\pi^{\gamma}}(b_0) = R(b_0, \pi^{\gamma}(b_0)) + \mathbb{E}\Big[\sum_{t=1}^{\infty} \gamma^t R(b_t, \pi^{\gamma}(b_t))\Big]$$

$$< R(b, \pi^{\gamma}(b_0)) + \mathbb{E}\Big[\sum_{t=1}^{\infty} R(b_t, \pi^{\gamma}(b_t))\Big]$$

$$\leq R(b, \pi^{\gamma}(b_0)) + \mathbb{E}\Big[\sum_{t=1}^{\infty} R(b_t, \pi^*(b_t))\Big] \leq P_{\mathcal{M}}^{\pi^*}(\Diamond T)$$

$$\square$$

Consider a problem instance such that the optimal policy requires $n$ steps to reach an optimal probability of $p$. Using $\gamma < 1$ gives an optimal value of $\gamma^n p$ in the worst case. Hence, discounting can lead to arbitrarily large under-approximations.

**Trials may not terminate for $\gamma = 1$.** When $\gamma < 1$, $\epsilon\gamma^{-t}$ is a strictly increasing and unbounded function, in which case algorithm HSVI2 is guaranteed to converge. However, when $\gamma = 1$, HSVI2 may not terminate for some POMDPs. Unsurprisingly, a similar phenomenon is shown to exist for Goal-POMDPs [Horák et al., 2018], which is also an indefinite horizon problem. Consider the belief MDP in Figure 1, with some initialized value function bounds (e.g., with the blind policy for the lower bound [Kochenderfer et al., 2022] and the $V_{MDP}$ method for the upper bound [Hauskrecht, 2000]). Starting from $b_1$, Eq. (3) chooses action $a$, and selects observation $o_1$ since it has the largest WEU, returning to $b_1$. The trial will thus be stuck at $b_1$ indefinitely, since the termination condition is never met.

**Loops.** In a POMDP, it is possible to choose actions and observations such that the same belief states are repeatedly reached, i.e., a loop. This is depicted in Figure 1, where taking action $a$ at $b_1$ and $b$ at $b_2$ is a policy that allows the agent to remain at $b_1$ and $b_2$. Without properly accounting for loops, the algorithm may either get stuck in a loop indefinitely during exploration, or can be otherwise ineffective due to repeatedly exploring the same sequences of beliefs.

In a discounted POMDP, the IE-MAX heuristic of HSVI2 performs well, since an action with the highest upper bound

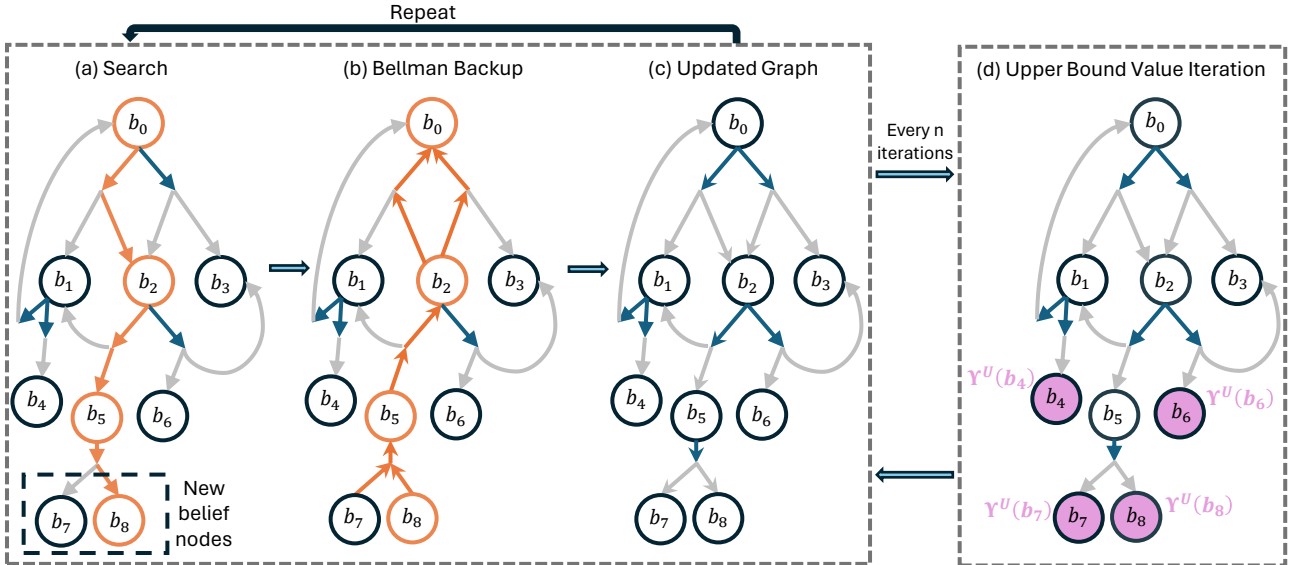

Figure 2: Overview of HSVI-RP. The algorithm incrementally constructs a belief *graph* using trial-based search, and maintains upper and lower bound values for each belief node. In each trial, actions and observations are selected using a heuristic based on the upper and lower bound values to visit a sequence of belief nodes (orange). After each trial, value bounds for each visited node are updated using local Bellman backups. Every $n$ iterations, upper bound values for all nodes are reset, and frontier nodes (purple) are re-initialized using the upper bound value set $\Upsilon^U$, and value iteration is performed. This allows better improvement of upper bound values for MRPP.

will be revealed to be suboptimal if its upper bound eventually decreases below the upper bound of another action. However, in MRPP, due to the presence of loops, many actions may have similar or the same upper bound values at a given belief. The IE-MAX heuristic may repeatedly choose the same actions and be stuck in a loop indefinitely, and new beliefs at the frontier may not be expanded to improve the upper bounds. For example, in the Belief MDP in Figure 1, from $b_3$, taking actions $a$ and $b$ has the same value, so $a$ may be chosen and $b_4$ is never expanded. Further, trial-based local Bellman backups for MRPP may not converge for upper bounds due to the presence of these so-called *end components*.[1] We discuss this in detail in Section 4.

## 4 HEURISTIC GRAPH SEARCH VALUE ITERATION

In this section, we present our algorithm, called Heuristic Search Value Iteration for Reachability Probabilities (HSVI-RP). The algorithm explores the search space by incrementally constructing a belief MDP graph $G$. The nodes in $G$ are allowed to have have multiple parents to alleviate the aforementioned issue of loops in trial-based search. The initial node of $G$ is the initial belief $b_0$. From $b_0$, the algorithm incrementally unrolls a finite fragment of the full

belief MDP through trial-based search, and maintains sound two-sided bounds on maximal reachability probability. The two-sided bounds are used to inform the direction of search, detect $\epsilon$-optimality, and bound the optimal solution when used in an anytime manner. These bounds also have the benefit that bound improvements in one part of the belief space also improve the bounds in other parts of the belief space.

HSVI-RP is outlined in Algorithm 1 and depicted in Figure 2. At each iteration, a depth-first trial is conducted. An action is heuristically selected at belief $b$, and all successor beliefs $b'$ and their transitions are added to the graph. To select the next belief in the trial, an observation is heuristically selected. When a trial is terminated, we perform Bellman backups over the selected belief nodes. We also perform exact value iteration on $G$ periodically to improve upper bounds. The key is to search the belief MDP efficiently by expanding belief nodes that may be part of the reachable space of an optimal policy. To do so, we maintain and update a set of upper and lower bounds. Each graph node has an associated upper and lower bound value computed from these sets. Then, we propose new search heuristics that take advantage of these bounds and the structure of Problem 1.

The algorithm can be seen as a modification of discounted POMDP trial-based search to address their drawbacks for MRPP. The following summarize our key modifications:

- We do not use discounting. Instead of terminating trials

---

[1]An end component is a sub-belief-MDP with a set of belief states $B' \subseteq B$ for which there exists a policy that enforces, from any state in $B'$, only the states in $B'$ are visited infinitely often.

**Algorithm 1** HSVI-RP $(\mathcal{M}, \epsilon, \mathrm{T})$

**Global**: $\mathcal{M}, G, V^L, \Upsilon^U$
1: Initialize $G$ with $b_0$
2: Initialize $V^L$ with blind policy
3: Initialize $V^U = \Upsilon^U$ with $V_{MDP}$
4: **while** $V^U(b_0) - V^L(b_0) > \epsilon$ **do**
5:     **for** n iterations **do**
6:         EXPLORE$(b_0, 0, V^U(b_0) - V^L(b_0), d_{\mathrm{trial}})$
7:     **end for**
8:     $\forall b \in G, V^U(b) = V^L(b)$
9:     $\forall b \in F, V^U(l) = \Upsilon^U(b)$
10:    Perform Value Iteration for Upper Bounds on belief MDP $G$ to obtain a new $\Upsilon^U$
11:    Increase $d_{\mathrm{trial}}$ if no improvements for $i$ iterations
12: **end while**
13: **return** $G, V^L, \Upsilon^U$

---

**Algorithm 2** EXPLORE$(b, t, \epsilon, d_{\mathrm{trial}})$

**Global**: $\mathcal{M}, G, \Upsilon^U, V^L, \kappa$
1: **if** $V^U(b) - V^L(b) \leq \kappa \cdot \epsilon$ **or** $t > d_{\mathrm{trial}}$ **then**
2:     **return**
3: **end if**
4: $A' \leftarrow \{a \ : \ \max_{a'} Q^U(b, a') - Q^U(b, a) < \xi\}$
5: $a^* \leftarrow \underset{a \in A'}{\mathrm{argmax}} \left[ Q^U(b, a) + c_a \sqrt{N(b)} / \left(1 + N(b, a)\right) \right]$
6: $o^* \leftarrow \underset{o}{\mathrm{argmax}} \left[ \mathrm{WEU}(b, t, \epsilon) + P(o|b, a^*) \frac{c_z \sqrt{N(b, a^*)}}{1 + N(b')} \right]$
7: Add all $b'$ from taking $a^*$ at $b$ to belief MDP graph $G$
8: Update $b_{t+1}$ using $a^*, o^*$ with Eq. (1)
9: EXPLORE$(b_{t+1}, t + 1, \epsilon, d_{\mathrm{trial}})$
10: Perform local updates on bounds $\Upsilon^U, V^L$ at belief $b$
11: **return**

---

with $\gamma^{-t}$, we use an adaptively increasing search depth to incrementally increase the depth of explored beliefs.

- We represent the search space as a graph by merging belief states that already exist in the graph.

- We propose new trial-based expansion heuristics to handle indefinite horizon graph search.

- To enable the improvability of two-sided bounds while maintaining tractability, we use a combination of local Bellman backups and exact value iteration on $G$.

Below, we provide details on the algorithm.

**Lower and Upper Bounds** An important reason for the effectiveness of trial-based search is the use of bounds that allow improvements in one part of the belief space to improve the bounds in other parts of the belief space. Thus, we utilize bound representations that have this property.

We use a set $\Gamma$ of $\alpha$-vectors for lower bound representation.

To initialize sound lower bounds, we use the blind policy [Kochenderfer et al., 2022] by taking some $i \geq 0$ number of steps, yielding lower bound on reachability probabilities. This set of $\alpha$-vectors also represents the policy for execution. Since $V^*(b) \geq V^\pi(b) = \max_{\alpha \in \Gamma}(\alpha^T b)$, the action at belief $b$ is chosen using $\arg\max_{\alpha \in \Gamma}(\alpha^T b)$.

We use a belief point set $\Upsilon^U$ to represent upper bounds. The upper bound value $V^U(b)$ at any belief $b$ is the projection of $b$ onto the convex hull formed by $\Upsilon^U$ of belief-value points $(b_i, V^U(b_i))$. We denote this projection as $\Upsilon^U(b)$, where $V^U(b) = \Upsilon^U(b)$. To initialize sound upper bounds, we use the $V_{MDP}$ method [Hauskrecht, 2000], which uses optimal values obtained on the fully observable underlying MDP. The MDP optimal value function provides values at the corners of the belief simplex, which are the initial points in the upper bound point set. These bounds can be further improved using the $Q_{MDP}$ or Fast Informed Bound methods. An upper bound for a belief is computed using an LP or a sawtooth approximation Hauskrecht [2000].

**Value Updates** The Bellman update equation allows us to update and improve the bounds through dynamic programming. The Bellman operator $\mathbb{B}$ is defined as:

$$Q(b, a) = R(b, a) + \mathbb{E}[V(b_{t+1})], \quad (8)$$
$$[\mathbb{B}V](b) = \max_a Q(b, a) \quad \forall b \in B. \quad (9)$$

$\mathbb{B}$ is defined over the entire belief space. For discounted POMDPs, it has been shown that performing an asynchronous local Bellman update over the belief states sampled in each trial is more efficient:

$$[\mathbb{B}V](b) = \max_a Q(b, a) \quad \forall b \in B_{trial} \subseteq B. \quad (10)$$

Our trial backup step performs asynchronous local Bellman update for both lower and upper bounds. Each application of $\mathbb{B}$ on a belief state adds an $\alpha$-vector to $\Gamma$ [Smith and Simmons, 2004], and updates the upper bound point set. As more of the belief space is explored during graph search, successive local updates leads to uniform improvement in the lower bounds and propagates improvements in the upper bound. Asynchronous local backups over trial sampled beliefs are effective for improving lower bounds.

**Periodic Exact Upper Bound Value Iteration** Unlike for discounted POMDPs, local backups over the upper bound point set may not lead to improvements of the upper bounds. Bellman backups over an upper bound for a POMDP may never improve because the Bellman operator for MRPP (which is reducible to the stochastic shortest path problem) is a semicontractive model [Bertsekas, 2022]. Consequently, there may be many fixed points for value function upper bound, with the optimal upper bound solution being the *least fixed point*, denoted by lfp[$V^U$].

Intuitively, this issue arises when there are loops, and thus states in end components may have upper bound values

higher than $\text{lfp}[V^U]$. This may cause a Bellman update to not decrease the upper bound value. Consider the example in Figure 1 again. If $b_4$ is added to the graph, and the upper bound value of $b_3$ is updated via Bellman backup to a value below 1, the backup for upper bounds at $b_2$ (and hence $b_1$) will not decrease in value because action $b$ gives the highest upper bound value of 1. When using local backups over the upper bound set, backups over belief states that are in an end component may not improve their upper bound values.

In finite state MDPs, when initialized with a suitable under-approximate value function, value iteration (VI) converges to $\text{lfp}[V]$ *from below* [Hartmanns and Kaminski, 2020]. By treating the frontier nodes of the partially explored belief MDP as an "upper bound target set" and via a suitable initialization, we can achieve a similar result for upper bounds for a given $G$.

Let $F$ be the set of frontier nodes of $G$. An upper bound on the maximal probability of reaching T by first going to a belief node $b \in F$ is

$$P_G^{\pi^*}(\lozenge\text{T}) = \max_\pi\{\mathbb{E}_{b\in F}[P_G^\pi(\lozenge b) + V^U(b)]\} \geq P_M^{\pi^*}(\lozenge\text{T}).$$

Thus, this reduces to computing maximal values, given upper bound values of the frontier nodes.

What remains is a suitable initialization. Intuitively, we want to fix $V^U$ for the frontier nodes, and under-approximate it for all the other nodes. Hence, for each $b \in F$, we set $V^U(b)$ to $\Upsilon^U(b)$. For all other nodes, the upper bound values are set to an under-approximation, such as their lower bounds, i.e., $V^U(n) = V^L(n)$. Then, VI of the upper bound values over $G$ until convergence obtains a new upper bound point set $\Upsilon^{U'}$, which is the least fixed point for $G$ and $\Upsilon^U$.

While VI is crucial to improve upper bounds, it has large computation overhead as VI has to be conducted for all nodes in the belief graph. Therefore, we periodically re-initialize the upper bound values to re-compute $\text{lfp}[V^U]$ over $G$ as more beliefs are added to $G$. This allows continual improvement of the upper bound over iterations by reusing the least fixed point from previous iterations instead of starting from a loose upper bound. As more of the belief space is expanded, the upper bound values improve towards $P_M^{\pi^*}(\lozenge\text{T})$.

**Trial-based Graph Exploration**  Here, we present a trial-based belief exploration technique modified for graphs in MRPP. We also propose a technique to handle loops.

*Action Selection:*  As discussed in Section 3, many actions may have the same upper bound value, so HSVI2's action selection method is ineffective. We propose an action selection heuristic based on the Upper Confidence Bound (UCB) [Coquelin and Munos, 2007], considering the upper bound $Q$ values plus a term based on the number of times that action has been selected. We additionally only consider actions that have upper bound values within some

user-specified action selection radius $\xi$ from the highest upper bound:

$$A' = \{a \;:\; |Q^U(b,a) - \arg\max_{a'} Q^U(b,a')| < \xi\}, \quad (11)$$

$$a^* = \arg\max_{a\in A'}\left[Q^U(b,a) + c_a\sqrt{N(b)}/(1 + N(b,a))\right]$$

where $c_a$ is an exploration constant, and $N(b)$ and $N(b,a)$ are the number of times $b$ has been visited, and action $a$ has been chosen at $b$ respectively. This heuristic incentivizes exploration of other actions that have similar upper bounds. A lower $\xi$ favor actions with higher upper bounds, improving upper bounds faster, but may reduce efficiency by limiting exploration.

*Observation Selection:*  Similarly, for observation selection, just using Eq. (5) is ineffective, as the same sequence of observations may be repeatedly chosen at a belief. Instead, we use a heuristic that is weighted based on the Excess Uncertainty, probability of reaching that observation, and number of times $N(b')$ the resulting belief has been chosen.

$$o^* \leftarrow \arg\max_o\left[\text{WEU}(b,t,\epsilon) + P(o|b,a^*)\frac{c_z\sqrt{N(b,a^*)}}{1 + N(b')}\right] \tag{12}$$

where $c_z$ is an exploration constant, and $N(b,a^*)$ is the number of times action $a^*$ has been chosen at node $b$. This heuristic incentivizes choosing successor belief states that have not been explored often.

Higher values of of $c_a$ and $c_z$ encourage more exploration but can hinder convergence if they are set too high due to too much exploration and too little exploitation.

**Remark 3** (Observation Heuristic Randomization)**.** *A heuristic that mixes between Eq.* (5) *and Eq.* (12) *also works well empirically. When randomization is used, with probability p, we randomize between using Eq.* (5) *and* (12).

Additionally, to address the problem of loops in graph search, we also keep track of the beliefs, actions, and observations that have been sampled during a trial, to not repeatedly choose the same sequences of actions and observations. When an action is selected, we only consider observations that do not lead to beliefs that are already sampled during the trial. Such beliefs are part of loops. If no observations are available from that action, we avoid selecting that action, and consider another action instead. If no more actions are available, we skip to the next belief in the sampled sequence that is part of the loop, and continue the trial. The trial ends if all beliefs have no actions available. Alternatively, one can maintain a global history list to not select the same histories more than once, but maintaining this history list may be ineffective if many histories end up in the same beliefs, and can be computationally demanding due to the number and length of histories in an indefinite horizon problem.

*Adaptive Trial Termination:* We define a maximum depth $d_{\text{trial}}$ for each trial, that is increased adaptively as the number of iterations increase. We use a simple heuristic to increase $d_{\text{trial}}$. $d_{\text{trial}}$ is increased by $d_{\text{inc}}$ when neither the bounds have changed by at least $0.01$ over $n$ successive trials. A trial terminates when either of two conditions hold: $t > d_{\text{trial}}$ or $V^U(b_t) - V^L(b_t) \leq \kappa \cdot (V^U(b_0) - V^L(b_0))$ for $0 < \kappa < 1$. Parameters $d_{trial}$ and $d_{inc}$ control the rate of increase of search depth. Higher values are beneficial for long horizon problems but may slow search efficiency if increased too quickly.

**Pruning** The size of the $\alpha$-vector set affects backups significantly. To keep the problem tractable as more of the search space is expanded, we prune dominated elements in the lower bound $\alpha$-vector set in a manner similar to HSVI2. $\alpha$-vectors are pruned when they are pointwise dominated by other $\alpha$-vectors. Pruning is conducted when the size of the set has increased by $10\%$ since the last pruning operation.

### THEORETICAL ANALYSIS

Here, we analyze the theoretical properties of HSVI-RP without observation heuristic randomization; specifically its soundness and asymptotic convergence.

Although our algorithm is inspired by the trial-based HSVI2, its properties are different due to the indefinite horizon property of MRPP and the modifications made to HSVI-RP. The proof for $\epsilon$-convergence of HSVI2 relies on discounting to bound the required trial depths and number of iterations. Additionally, loops are not an issue due to discounting. On the other hand, HSVI-RP's asymptotic convergence of the lower bound for MRPP stems from our proposed graph representation, termination criteria, and trial-based expansion heuristics, which allow adequate exploration of the belief MDP.

**Lemma 1** (Soundness). *At any iteration of HSVI-RP, it holds for all $b_t \in B$ that $V^L(b_t) \leq V^*(b_t) \leq V^U(b_t)$.*

**Theorem 1** (Asymptotic Convergence). *Let action selection radius in Eq. (11) be $\xi = 1$. Further, let $V_n^L$ denote the lower bound obtained from HSVI-RP after trial iteration $n$. Assume that there exists an optimal belief-based policy $\pi^*$ computable with finite memory. Then,*

$$\lim_{n \to \infty} \left[ P_{\mathcal{M}}^{\pi^*}(\lozenge \mathrm{T}) - V_n^L(b_0) \right] = 0.$$

Proofs of Lemma 1 and Theorem 1 can be found in the Appendix.

In general, optimal belief-based policies for POMDPs with indefinite horizon require infinite memory, and the corresponding decision problem is undecidable [Madani et al., 2003]. Therefore, the finite memory assumption is a practical requirement for any computed policy.

We leave the complete analysis of convergence of the upper bound to future work. It is possible to guarantee upper bound convergence in certain cases, e.g., when the POMDP induces a finite belief MDP. However, there are MRPPs in which the upper bound does not converge. These are MRPPs where lowering the upper bound requires an infinite number of explored beliefs, related to the undecidability result for indefinite horizon POMDPs. In our experimental evaluations, we found that the upper bound converges (quickly) in some problems but not in others.

## 5 EMPIRICAL EVALUATION

In this section, we evaluate our proposed algorithm. We aim to answer the following questions in our evaluation.

**Q1.** *How well do discounted trial-based POMDP algorithms perform for MRPP?* We study the effect of discounting on solution quality for indefinite horizon reachability. We use SARSOP Kurniawati et al. [2009] together with Bouton et al. [2020], with varying levels of discount factor.

**Q2.** *How does our approach compare to state-of-the-art belief-based approaches?* We compare our approach to those by PRISM Norman et al. [2017], STORM Bork et al. [2022] and Overapp Bork et al. [2020]. PRISM computes two-sided bounds using a grid-based discretization to approximate the belief MDP. STORM and Overapp compute lower and upper bounds, respectively, in a breadth first search manner.

**Q3.** *How does our approach compare to other state-of-the-art approaches?* We compare our approach to PAYNT Andriushchenko et al. [2022] and SAYNT Andriushchenko et al. [2023]. PAYNT is an inductive synthesis algorithm that searches in the space of (small-memory) FSCs. SAYNT is an algorithm that integrates both PAYNT and STORM by using the FSCs computed from one to improve the other in an anytime loop.

**Benchmarks and Setup** We implemented a prototype of HSVI-RP in Julia under the POMDPs.jl framework [Egorov et al., 2017], and used the available open source toolboxes for the other algorithms. We use benchmark MRPPs from [Bork et al., 2022], with variants on size and difficulty. Details on the problems can be found in the Appendix. For HSVI-RP, observation heuristic randomization (Remark 3) with $p = 0.5$ was used for the Drone problems, and we report the mean of the bounds obtained from 10 runs. All experiments were run on a single core of a machine equipped with an Intel i7-11700K @ 3.60GHz CPU and 32 GB of RAM. Our code is open sourced and available on GitHub[2].

---

Table 1: Performance of discounted-sum SARSOP. Bold indicates best results, and '$-$' indicates not converged at timeout ($900s$).

| | SARSOP | | | | | | Ours |
|---|---|---|---|---|---|---|---|
| | $\gamma = 0.95$ | $\gamma = 0.98$ | $\gamma = 0.99$ | $\gamma = 0.999$ | $\gamma = 0.99999$ | $\gamma = 1 - 10^{-16}$ | |
| Grid-av 4 | [0.758, 0.758]
$< 1s$ | [0.857, 0.857]
$< 1s$ | [0.892, 0.892]
$< 1s$ | [0.923, 0.923]
$< 1s$ | **[0.928, 0.928]**
**< 1s** | **[0.928, 0.928]**
**< 1s** | [0.928, 0.928]
**< 1s** |
| Grid-av 20 | [0.028, 0.049]
$-$ | [0.155, 0.212]
$-$ | [0.332, 0.38]
$-$ | [0.709, 0.721]
$-$ | [0.781, 0.782]
$< 1s$ | [0, 1]
$-$ | **[0.782, 0.783]**
**33s** |
| Refuel6 | [0.21, 0.21]
$< 1s$ | [0.32, 0.33]
$< 1s$ | [0.39, 0.39]
$3.6s$ | [0.63, 0.63]
$200s$ | [0.20, 0.98]
$-$ | [0.18, 0.98]
$-$ | **[0.67, 0.67]**
**1.4s** |
| Refuel8 | [0.184, 0.184]
$1.4s$ | [0.314, 0.314]
$4.24s$ | [0.374, 0.375]
$9.83s$ | [0.438, 0.439]
$339s$ | [0.218, 0.987]
$-$ | [0.00, 0.988]
$-$ | **[0.445, 0.445]**
**20s** |

Table 2: Results for benchmark POMDPs. The top entry refers to the computed values, and bottom entry (left) is the time taken to achieve that value. For belief-based approaches (PRISM, STORM, Overapp, Ours), we additionally report the number of beliefs on the bottom entry (right). Bold entries denote the best solutions (best value, followed by lowest runtime if multiple methods achieve the same value). TO/MO denotes timeout (2 hours) or out of memory with no solution, and '$*$' indicates that the reported result is from previous papers.

| | PRISM | STORM | PAYNT | SAYNT | **Ours** | Overapp |
|---|---|---|---|---|---|---|
| Nrp8 | **[0.125**, 0.189]
$58s$, 735K beliefs | $\geq$ **0.125**
$< 1s$, 50 beliefs | $\geq$ **0.125**
$< 1s$ | $\geq$ **0.125**
$< 1s$ | **[0.125, 0.125]**
**< 1s, 32 beliefs** | $\leq$ **0.125**
$< 1s$, 50 beliefs |
| Crypt4 | **[0.33**, 0.77]
$33s$, $312K$ beliefs | $\geq$ **0.33**
**< 1s, 560 beliefs** | $\geq$ **0.33**
$< 1s$ | $\geq$ **0.33**
$< 1s$ | **[0.33, 0.33]**
$15.6s$, 480 beliefs | $\leq$ **0.33**
**< 1s, 560 beliefs** |
| Rocks12 | TO/MO | 0.63
$1223s$, 2M beliefs | $\geq$ **0.75**
$5.7s$ | $\geq$ **0.75**
$5.6s$ | **0.75, 0.75**
**2.8s, 770 beliefs** | $\leq$ **0.75**
**< 1s, 2.5K beliefs** |
| Grid-av 4 | [0.21, 1.0]
$< 1s$, 15 beliefs | $\geq$ **0.928**
$118s$, 10M beliefs | $\geq$ **0.928**
$158s$ | $\geq$ **0.928**
$87.2s$ | **[0.928, 0.928]**
**< 1s, 194 beliefs** | $\leq 0.984*$
$6.3s$, 125K beliefs |
| Grid-av 10 | [0, 0.999]
$34s$, 97 beliefs | $\geq$ 0.513
$75s$, 5M beliefs | $\geq$ 0.744
$309s$ | $\geq$ **0.773**
$337s$ | **[0.773, 0.774]**
**8s, 8K beliefs** | $\leq$ 1.0
$< 1s$, 5K beliefs |
| Grid-av 20 | TO/MO | $\geq$ 0.115
$77s$, 5M beliefs | $\geq$ 0.524
$1156s$ | $\geq$ 0.667
$1986s$ | **0.782, 0.783**
**33s, 12K beliefs** | $\leq$ 1.0
$1.2s$, $75K$ beliefs |
| Drone 4-1 | TO/MO | $\geq$ 0.839
$210s$, 6.5M beliefs | $\geq$ 0.869
$2509s$ | $\geq$ **0.890**
**427s** | [0.884, 0.957]
$7200$, $51K$ beliefs | $\leq$ **0.942**$*$
$1270s$, 14M beliefs |
| Drone 4-2 | TO/MO | $\geq$ 0.953
$207s$, 8.8M beliefs | $\geq$ 0.963
$6815s$ | $\geq$ **0.971**
**733s** | [0.964, 0.976]
$7200s$, 26K beliefs | $\leq$ **0.974**
**44s, 762K beliefs** |
| Refuel6 | **[0.67**, 0.72]$*$
$136s$, 6K beliefs | $\geq$ **0.672**
$1.4s$, 4.5K beliefs | $\geq$ **0.672**$*$
$77.8s$ | $\geq$ **0.672**
$85s$ | **[0.672, 0.672]**
**1.4s, 387 beliefs** | $\leq$ 0.687
$< 1s$, 48K beliefs |
| Refuel8 | TO/MO | $\geq$ 0.439
$1.7s$, 20K beliefs | $\geq$ **0.445**
$494s$ | $\geq$ **0.445**
$91s$ | **[0.445, 0.446]**
**20s, 3.7K beliefs** | $\leq 0.509*$
$410s$, 11M beliefs |
| Refuel20 | TO/MO | $\geq$ 0.144
$142s$, 3.9M beliefs | $\geq$ 0.018
$1666s$ | $\geq$ 0.204
$937s$ | **[0.328, 0.999]**
**1356s, 32K beliefs** | $\leq$ **0.999**
**7.56s, 177K beliefs** |

**Q1: Performance of trial-based discounted-sum algorithms.** Table 1 summarizes the key results to answer Q1. From Table 1, we see that there is not a clear way to use a discounted POMDP algorithm to get good performance for indefinite horizon maximal reachability probabilities problems. A typical discount factor used in discounted-sum POMDP problems is $0.95$ to $0.999$. Unsurprisingly, such values of discounting under-estimate the optimal probabili-

ties. Therefore, one should increase the discount factor to as close to 1 as possible, to get the best probabilities. However, for problems with more loops in the belief transitions, such as Refuel6 and Refuel8, increasing the discount factor causes trials to be deeper but search may not be effective (or trials may not terminate). In all cases, HSVI-RP performs as well or better than using discounted SARSOP directly.

Table 2 reports a set of benchmarks for algorithms that

do not use discounting[3] to answer Q2-Q3. PRISM and HSVI-RP provide two-sided bounds. STORM, PAYNT and SAYNT provide under-approximations, while Overapp provides over-approximations. We also provide plots of the evolution of our value bounds over time in the Appendix. We report the computed values, time taken to achieve that value, and also reports the number of beliefs expanded for the compared belief-based approaches (PRISM, STORM, Overapp, and HSVI-RP). Note that PAYNT does not expand beliefs, and although SAYNT expands beliefs, it does not report the number of beliefs expanded. As reported in [Andriushchenko et al., 2023], SAYNT typically reduces the memory usage of STORM by a factor of 3-4.

**Q2: Comparison to belief expansion-based approaches.** HSVI-RP generally performs better than other belief-based approaches, exceeding the accuracy of their under- and over-approximations with faster convergence for most of the problems. Additionally, HSVI-RP expands orders of magnitude fewer beliefs than STORM and Overapp, requiring less memory. All three compared algorithms were not able to improve their computed values much more than their best solutions due to the memory intensity of grid-based approximations and breadth-first belief expansion, while HSVI-RP was able to improve computed policies over time due to the trial-based methodology. These results strongly suggest that depth-first trial-based heuristic search that utilizes $\alpha$-vector and upper bound point set representations, is an effective belief expansion methodology for MRPP.

**Q3: Comparison to other approaches.** HSVI-RP is highly competitive compared to both PAYNT and SAYNT, achieving better lower bounds in many of the problems, while also providing upper bounds. Further, HSVI-RP generally (except in the Drone problems) finds optimal solutions faster than both methods. For Refuel6, Refuel8, Grid-av 4, and Grid-av 10, SAYNT computed near-optimal solution within 377s, but continued computations until timeout without detecting it. On the other hand, both of HSVI-RP's bounds converged for these problems, allowing termination with a near-optimal policy. Hence, the use of two-sided bounds can help inform when a near-optimal policy is found.

**Further Discussion.** Although the efficiency of our approach is promising, we note that the rate of convergence towards the optimal probabilities can be slow for larger problems which require deep trials. From preliminary analysis, the computation time is mainly bottlenecked by Exact Upper Bound Value Iteration and Bellman backups over large numbers of $\alpha$-vectors during deep trials. Heuristic search using two-sided bounds is also less effective when the upper

bound values are uninformative (as is the case for the Drone problems), possibly due to the presence of large ECs. Better model checking techniques, such as on-the-fly detection and handling of ECs, may improve belief exploration and convergence.

Similar to STORM, HSVI-RP benefits greatly from seeding with a good policy. Our policy initialization is the blind policy, which achieves a lower bound of 0 for most problems. In contrast, SAYNT's inductive synthesis approach finds good initial policies quickly. SAYNT performs the best among these algorithms by leveraging the strengths of the belief exploration of STORM and FSC generation of PAYNT. An integrated approach with HSVI-RP and PAYNT may be a good direction for scalable and fast verification and policy synthesis.

# 6 CONCLUSION

This paper studies the problem of computing near-optimal policies with both lower and upper bounds for POMDP MRPP. Utilizing ideas from heuristic trial-based belief exploration for discounted POMDPs, we propose an incremental graph search algorithm that searches in the space of potentially optimal reachable beliefs. This work shows that trial-based belief exploration can be an effective methodology to obtain near-optimal policies with over- and under-approximations of reachability probabilities.

### Acknowledgements

This work was supported by Strategic University Research Partnership (SURP) grants from the NASA Jet Propulsion Laboratory (JPL) (RSA 1688009 and 1704147). Part of this research was carried out at JPL, California Institute of Technology, under a contract with the National Aeronautics and Space Administration (80NM0018D0004).

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

# Sound Heuristic Search Value Iteration for Undiscounted POMDPs with Reachability Objectives
# (Supplementary Material)

**Qi Heng Ho**[1]     **Martin S. Feather**[2]     **Federico Rossi**[2]     **Zachary Sunberg**[1]     **Morteza Lahijanian**[1]

[1]Department of Aerospace Engineering Sciences, University of Colorado Boulder, Boulder, Colorado, USA
[2]Jet Propulsion Laboratory, California Institute of Technology, Pasadena, California, USA

## A   PROOF OF LEMMA 1

*Proof.* We first show that the initial upper and lower bounds are sound.

We initialize the lower bound $\alpha$-vectors with a blind policy for $i$ steps, which is a lower bound on the maximal reachability probability. Upper bounds are initialized with the $V_{MDP}$ technique, which assumes that there will be full observability after taking the first step. Since we can only do better if we have full observability, the computed value function is an upper bound on the optimal value function [Kochenderfer et al., 2022]. Therefore, the initialized bounds are sound.

Now, we show that iterations of belief exploration and backups preserve soundness of the bounds.

An $\alpha$-vector obtained from a Bellman backup of an $\alpha$-vector set is proven to remain a lower bound as long as the $\alpha$-vector set is a lower bound [Kochenderfer et al., 2022].

A Bellman backup of an upper bound point $V^U$ is:

$$V^{U'}(b) = \max_{a \in A} \left( \mathbb{E}(R(s,a)) + \sum_o Pr(o|b,a)V^U(\tau(b,a,o)) \right) \geq \max_{a \in A} \left( \mathbb{E}(R(s,a)) + \sum_o Pr(o|b,a)V^*(\tau(b,a,o)) \right) = V^*(b)$$

i.e., the upper bound point remains an upper bound.

Finally, we show that Exact Upper Bound Value Iteration preserves the upper bound property of the upper bound point set. Let $V^U(l)$ be an upper bound maximal probability of reaching T from $l \in L$ given $G$. The Exact Upper Bound Value Iteration computes the upper bound on the maximal probability of reaching T by first going to a node $l \in L$:

$$P_G^{\pi^*}(\lozenge T) = \max_\pi \{ \mathbb{E}_{l \in L}[P_G^\pi(\lozenge l) + V^U(l)] \} \geq P_M^{\pi^*}(\lozenge T).$$

Since Exact Upper Bound Value Iteration is initialized with a sound upper bound $\Upsilon_i^U$ at frontier nodes, convergence of value iteration implies that the new upper bound point set $\Upsilon_{i+1}^U$ is also an upper bound. $\square$

## B   PROOF OF THEOREM 1

*Proof.* Let the current trial depth be $t_{trial} = d > 1$. We show that our algorithm eventually expands all beliefs reachable within $d$ steps.

Consider belief $b_0$ at the root of the graph, which will always be selected during a trial. From the action selection method of Eq. (11), all actions are eventually selected infinitely often as $n \to \infty$ since the second term $c_a \cdot \frac{\sqrt{N(b)}}{1+N(b,a)}$ is a strictly increasing function if $a$ is not selected. The observation selection method of Eq. (12) behaves in a similar manner, where $c_o \cdot \frac{\sqrt{N(b)}}{1+N(b_{t+1})}$ is a strictly increasing function if observation $o$ is not selected. Therefore, all beliefs will be selected infinitely often as $n \to \infty$. Therefore, all beliefs reachable within 1 step (depth 1) will eventually be expanded.

Next, during a trial, when a depth 1 belief is reached, all actions and observations are again eventually selected infinitely often as $n \to \infty$. By induction, all beliefs reachable within $d$ steps will eventually be expanded.

Suppose that the algorithm has searched all beliefs reachable within $d$ steps, and constructed the belief MDP $G_d$. Note that since belief MDP $G_d$ is a graph, policies over $G_d$ do not only include $d$-step trajectories, but also indefinite-horizon policies. Let $V_{G_d}^*(b_0)$ be the optimal value function for the belief MDP $G_d$ for Problem 1, i.e., maximal probability of reaching $s \in T$ *only within $G_d$*. Thus,

$$V_{G_d}^*(b_0) \leq V^*(b_0),$$

since there may be $s \in \mathrm{T}$ reachable from $b_0$ that are not in $G_d$ (reachable within $d$ steps).

Let $V_{0,\emptyset}^U, V_{0,\emptyset}^L$ be the upper and lower bounds at the initial iteration, and $V_{n,G_d}^U, V_{n,G_d}^L$ be the upper and lower bound fixed points computable with the belief MDP $G_d$ at iteration $n$, i.e., $V_{n,G_d}^L$ has converged to its fixed point (using asynchronous local updates) and $V_{n,G_d}^U$ has converged to its least fixed point (using Exact Upper Bound Value Iteration). At $b_0$, $V_{n,G_d}^L(b_0)$ upper bounds $V_{G_d}^*$, since the $\alpha$-vectors represent conditional plans that include the probability of reaching $s \in \mathrm{T}$ that are not in $G_d$,

$$V_{G_d}^*(b_0) \leq V_{n,G_d}^L(b_0) \leq V^*(b_0) \leq V_{n,G_d}^U(b_0),$$

Also, $d_{\mathrm{trial}} \to \infty$ as $n \to \infty$. Assume that an optimal policy can be represented with a finite $N$-memory belief-based policy. Since our states, actions and observations are finite, this implies that there exists $M \geq N$ where there is a trial depth $t_{trial} = M$ such that for a finite sized belief MDP $G_M$

$$V_{G_M}^*(b_0) = V^*(b_0) \implies V_{n,G_M}^L(b_0) = V^*(b_0)$$

Therefore,

$$\lim_{n \to \infty} |V^*(b_0) - V_n^L(b_0)| = 0$$

$\square$

The assumption that there exists a finite-memory is consistent with the results that the decision problem for POMDPs is undecidable, even in the discounted-sum case. Additionally, note that this does not only hold for POMDPS with a a finite reachable belief space, only that a finite belief MDP is sufficient to compute an optimal policy.

We remark that this proof considers the worst-case convergence of the algorithm, in which all beliefs and trajectories are expanded in an unbounded manner to reach an optimal solution. In practice, we can get near-optimal policies without needing to expand all possible nodes.

## C BENCHMARK PROBLEMS

**Nrp8** This problem is a non-repudiation protocol for information transfer, introduced as a discrete-time POMDP model by [Norman et al., 2017]. The goal is to compute the maximum probability, of a malicious behavior, that a recipient $R$ is able to gain an unfair advantage by obtaining information from an originator $O$ while denying participating in the information transfer.

**Crypt4** This problem models the the dining cryptographers protocol as a POMDP [Norman et al., 2017]. A group of N cryptographers are having dinner at a restaurant. The bill has to be paid anonymously: one of the cryptographers might be paying for the dinner, or it might be their master. The cryptographers respect each other's privacy, but would like to know if the master is paying for dinner. The goal is to know if the cryptographer's master is . See [Norman et al., 2017] for more details.

**Rocks12** The rock sample problem was considered for model checking by [Bouton et al., 2020]. It models a rover exploring a planet, tasked with collecting rocks. However, the rocks can be either `good` or `bad` and their status is not directly observable. The robot is equipped with a long range sensor, but sensing rock states is noisy. The problem ends when the robot reaches an exit area, with the state labelled as `exit`. We consider the formula $\phi_2 = \Diamond\texttt{good} \land \Diamond\texttt{exit}$ from [Bouton et al., 2020].

**Grid Avoid**  This is a classical POMDP problem introduced as a benchmark problem for MRPP by [Norman et al., 2017]. There is 1 obstacle and 1 target state in a $4 \times 4$ grid, and the goal is to reach the target state while avoiding the obstacle. In Grid-av 4-0.1, the agent has a $0.1$ probability of staying still when attempting to move to another grid. The agent has an initial belief distribution of being in any of the non-obstacle or target states. We extend the problem to a $10 \times 10$ grid with 3 obstacles in Grid-av 10-0.3, and the probability of staying still when attempting to move is increased to $0.3$. In Grid-av 20-0.5, there are 5 obstacles and the probability of staying still is increased to $0.5$. In both Grid-10-0.4 and Grid-10-0.5, the agent has initial belief distribution of being in any of the non-obstacle states within the first $5 \times 5$ grid.

**Drone**  In Drone N-R, the agent has to reach a target state in an $N \times N$ grid, while avoiding a stochastically moving obstacle. The obstacle is only visible within a limited radius $R$ [Bork et al., 2020].

**Refuel**  In RefuelN, the agent goal is to reach a target state in an $N \times N$ grid. There is uncertainty in movement and its own position is not directly observable. There are static obstacles, and movement requires energy. The agent starts with $N - 2$ energy, and each move action uses 1 energy. Energy can be refilled at recharging stations.

| Model | States | State-action pairs | Observations |
|---|---|---|---|
| Nrp8 | 125 | 161 | 41 |
| Crypt4 | 1972 | 4612 | 510 |
| Rocks12 | 6553 | $3 \cdot 10^4$ | 1645 |
| Grid-av 4-0.1 | 17 | 62 | 3 |
| Grid-av 10-0.3 | 101 | 389 | 3 |
| Grid-av 20-0.5 | 401 | 1580 | 3 |
| Drone 4-1 | 1226 | 3026 | 384 |
| Drone 4-2 | 1226 | 3026 | 761 |
| Refuel-06 | 208 | 565 | 50 |
| Refuel-08 | 470 | 1431 | 66 |
| Refuel-20 | 6834 | 25k | 174 |

Table 3: Size of Benchmark Problems

# D   ALGORITHM DETAILS AND PARAMETERS

**Discounted-Sum POMDP**  We used the technique in [Bouton et al., 2020] together with SARSOP [Kurniawati et al., 2009] (toolbox implementation in C++) to compute solutions.

**PRISM**  We used the toolbox implemented by [Norman et al., 2017]. We varied the parameter *resolution* and report the best results.

**Overapp**  We used the implementation in [Bork et al., 2020] in the toolbox STORM. We report the best results over the recommended parameters found in the paper.

**STORM, PAYNT, SAYNT**  We used the implementation of all three algorithms from the toolbox available in [Andriushchenko et al., 2023]. This toolbox is implemented in C++. The STORM implementation has multiple parameter settings - cut-off, clip2, clip4, or expanding {2, 5, 10, 20} million belief states. We report the best results for each experiments from these parameters. We use the default parameters for PAYNT, which method searches in the space of increasing $k$-memory FSCs. We report the best result. We report the results using the parameters recommended in the toolbox for SAYNT. SAYNT outputs two values (one for STORM and one for PAYNT); we report the best value. Overall, the results are similar to those in the original publication of these algorithms.

**HSVI-RP**  We use $c_a = 0.01$, $\xi = 0.1$, $c_z = 0.01$, initial $d_{\text{trial}} = 200$, $d_{\text{inc}} = 10$, $\kappa = 0.01$, and performed Exact Upper Bound Value Iteration every 10 exploration trials for all experiments. For all experiments except Drone 4-1 and Drone 4-2, we used our proposed heuristic, so there is no randomization in the algorithm. For Drone 4-1 and Drone 4-2, we randomized (probability 0.5) between our proposed heuristic and the original HSVI2 heuristic, and report the mean results over 10 runs.

The benchmark problems have specifications in the form of co-safe LTL. We use the technique by Bouton et al. to compute product automata and accepting product target states.

**Evaluation Validity**    This evaluation focuses on the potential for trial-based search to obtain policies with tight two-sided bounds for maximal reachabiltiy probabilities through comparisons with state-of-the-art methods. It is important to note that some of the algorithms are implemented in different toolboxes and programming languages with varying levels of code optimization. To mitigate some of the issues related to this, we conducted all experiments on the same CPU when possible. Nonetheless, we do not draw definite conclusions on the relative speed of each algorithm due to their implementation differences.

# E    CONVERGENCE PLOTS

Figure 3 plots the evolution of our two-sided bounds for the evaluated benchmarks. The dashed and dotted lines give the other algorithms' final results for comparison. We omit the bounds obtained by PRISM as they are largely uninformative.

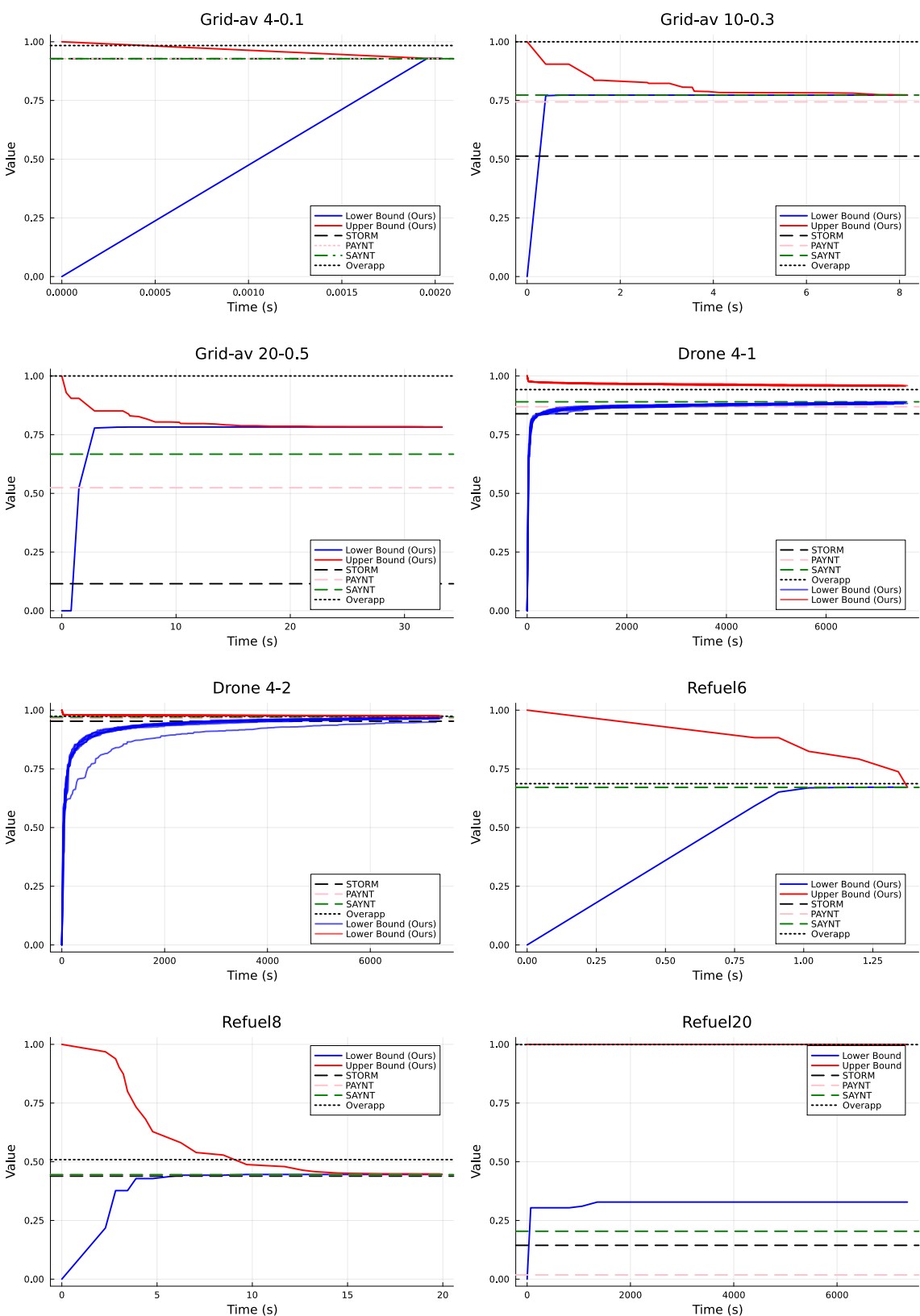

Figure 3: Evolution of lower and upper bound values over time. Overapp computes upper bounds, while STORM, PAYNT, and SAYNT compute lower bounds.