# OpenReview forum: "Sound Heuristic Search Value Iteration for Undiscounted POMDPs with Reachability Objectives"
_auai.org/UAI/2024/Conference — UAI 2024 poster_

### Official Review · Reviewer_MYHW · 2024-03-19

**Q2-1 Originality-Novelty:** 3
**Q2-2 Correctness-Technical Quality:** 3
**Q2-5 Clarity Of Writing:** 2

**Q1 Summary And Contributions:**

The paper presents the algorithm HSVI-RP to tackle the maximal reachability probability problem (MRPP) in partially observable Markov decision processes (POMDPs). The algorithm is an extension of existing so-called trial-based tree search methods typically used for discounted-sum objectives.

The main idea behind these methods is to explore the belief space underlying the POMDP using repeated trials of trajectories in the belief space. The authors point out problems with these approaches that arise when considering the (undiscounted) MRPP. In particular, theoretical guarantees about the quality of results and termination of the algorithms do not carry over from the discounted setting. HSVI-RP uses a graph-based representation of the search space as well as a specific termination criterion for trajectory trials to address these problems. Furthermore, they present new exploration heuristics and a combined backup strategy for upper and lower bounds.

HSVI-RP tracks both upper and lower bounds on the maximal reachability probability. The authors show that the bounds are sound and that the algorithm converges under the assumption that there exists a finite memory policy that optimises the reachability probability. A comparison of the implemented algorithm with the belief-based approaches in the tools PRISM and Storm as well as the inductive synthesis tool PAYNT show promising performance.

**Q2-3 Extent To Which Claims Are Supported By Evidence:**

2: Fair: the main claims are somewhat supported by evidence (but the experimental evaluation may be weak, or does not match entirely with the claims, important baselines may be missing, proofs contain important ideas but lack rigor, algorithmic details are only discussed superficially, references are imprecise, assumptions are not sufficiently motivated or explicated, etc.).

**Q2-4 Reproducibility:**

3: Good: key resources (e.g. proofs, code, data) are available and key details (e.g. proofs, experimental setup) are sufficiently well-described for competent researchers to confidently reproduce the main results.

**Q3 Main Strengths:**

- the practical performance of HSVI-RP compares well to existing methods
- the extension of point-based methods for POMDPs to the special case of maximal reachability probabilities is novel
- modifications to the original algorithm are well motivated and described quite detailed
- a good overview of related work is provided

**Q4 Main Weakness:**

- it is stated that the over-approximation converges to the optimal solution, however the proof is left for future work
- the assumption that there exists an optimal policy only occurs in Theorem 1, though the paper is written as if the assumption implicitly holds (in definitions etc.). In addition, I believe some of the benchmarks used do not confirm to this assumption.
- the extension to expected reward specifications is hinted at, though details are missing
- some details of the experimental setup for the evaluation are only part of the supplementary material (tool configurations)

**Q5 Detailed Comments To The Authors:**

- the claim that it is possible to extend HSVI-RP to reward structures (Section 5) needs more explanation. I do not think that it is easy to see how this is supposed to work.
- Section 1: "In addition, they [other methods] suffer from scalability" may need a reference
- Section 1, related work: I think the over-approximation in [Bork et al. 2020] is more based on the exploration of a discretisation of the belief space rather than cut-offs.
- in Table 2, there is a bit of inconsistency how runtimes <1s are reported (see Grid-av 4-0.1 "ours" vs. Refuel6 „Overapp“)
- in Table 2, it is not clear to me what exactly "*" indicates as both over- and under-approximation values are marked
- The distinction between „STORM“ and „Overapp“ in Section 5 hides the fact that both are part of Storm. This should be more clearly stated in the paper.

Spelling/Grammar:
- related work: "Belief's not added are cut-off, and values from a pre-computed policy *are* used"
- footnote 1: "sub-blief-MDP" should be "sub-belief-MDP"

Questions:
- what is the idea of the extension to rewards (see first comment)?
- the runtime reported for STORM on benchmark Refuel6 seems unexpectedly long (similar runtime to models where a lot more states have been explored). Is there an explanation for that?

**Q9 Complying With Reviewing Instructions:**

Yes

---

> ### Author Rebuttal · Authors · 2024-04-05
>
> We thank the reviewer for the thoughtful comments and questions.
>
> **Q4.1: Over-approximation convergence.**
>
> We'd like to clarify that we do not claim HSVI-RP has upper bound convergence guarantees, and we apologize for the confusion.  In the paper, we say that we empirically observe convergence for some of the benchmark problems.  We can guarantee upper bound convergence in certain cases, e.g., when the POMDP induces a finite belief MDP. But, there are MRPPs in which the upper bound does not converge. These are MRPPs where lowering the upper bound requires an infinite number of explored beliefs, related to the undecidability of infinite horizon POMDPs (Madini, Hanks, and Condon, AIJ'03). We will add this discussion and ensure clarity in the final version.
>
> We also remark that, for MRPP, there is a gap in the literature for a well-performing algorithm with two-sided convergence. Among the compared algorithms, only PRISM provides guarantees on asymptotic convergence; the other algorithms have provably sound bounds. Most of the algorithms provide one-sided bounds with no algorithm utilizing both bounds to guide search. HSVI-RP is a step in this direction with provably sound two-sided bounds used to guide the search. HSVI-RP has proven convergence of the lower bound, and achieves convergence of both bounds for some problems.
>
> **Q4.2: Assumption of existence of optimal finite memory policy.**
>
> We should clarify that the assumption is for an optimal finite memory policy. In a POMDP, it is known that the best memoryless policy (mapping observations to actions) can be arbitarily suboptimal in the worst case (Littman ICSAB'94). HSVI-RP finds belief-based policies, which are history-dependent. However, in general optimal belief-based policies for POMDPs with indefinite horizon require infinite memory, and is undecidable (Madini, Hanks, and Condon, AIJ'03). Thus, the finite memory assumption is a practical requirement for any computed policy.
>
> Note that for all benchmark problems except Drone, HSVI-RP finds policies with upper and lower bounds almost converged to a single value, indicating optimality.
> It is unclear if the Drone problem has an optimal finite memory policy, but we show that our method still performs well compared to existing methods in it, and monotonically improves over time, as seen in Figure 2 in Appendix E. To reduce subjectivity, we will reword "mild conditions" to "some conditions".
>
> **Q4.4: Experimental setup in Appendix.**
>
> We'll add these details into the main text.
>
> **Q5.Question1: What is the idea of the extension to rewards?**
>
> For clarity, we do not claim that HSVI-RP can be directly extended to other reward structures (e.g. maximizing a non-negative reward). The idea is to tackle the problem as a stochastic shortest path problem. Our lower bounds directly work for maximizing non-negative rewards. The main difficulty is in initializing upper bounds. The direction we are considering is problems where it is reasonable to restrict ourselves to proper policies (Bertsekas TAC'18). With proper policies, we can use the $V_{MDP}$ upper bound from an MDP. For future work, it may be possible to extend HSVI-RP to handle such reward structures under certain conditions such as proper policies.
>
> **Q5.Question2: Runtime for STORM on Refuel6.**
>
> We apologize for the confusion. There was a typo for Refuel6, where the time taken for STORM (should be 1.4s) and PAYNT (should be 77.8s) are swapped. These results are in line with the results reported in the original papers. We'll fix it in the final version. Nonetheless, this occurs in STORM in the benchmark Rocks12, shown in our additional experiments for Q1. In Rocks12, the target state is partially observable, which violates an assumption made in STORM. This may affect the heuristics used in STORM.
>
> **Q5.Comment2: "In addition, they [other methods] suffer from scalability" may need a reference**
>
> Thanks for pointing this out!  This has been our observation (specifically for STORM and PAYNT), but we shouldn't state it formally. For that reason, we will remove that statement.
>
> **Q5.Comment3: On Bork et al. (2020) method**
>
> Both the breadth-first exploration of a discretized belief space and cut-offs are used. Cut-offs are used to circumvent needing to explore the entire belief MDP.
>
> **Q5.Comment5: Table 2 - What does * indicate?**
>
>  "*" indicates that we used the best reported results from the cited paper, as we achieved significantly worse results over the parameters when ran on our machine. For the results without a *, we reproduced similar results to those in the cited papers. We will clarify it in the final version.
>
> **Q5.Comment6: STORM and Overapp are both part of Storm.**
>
> Thanks for this comment. Yes, they are both in the STORM toolbox, and there is an option to obtain both bounds sequentially. However, as they are not part of the same algorithm, the bounds are not used to inform each other, unlike this work. We will clearly state this in the paper.

---

### Official Review · Reviewer_3Pxw · 2024-03-22

**Q2-1 Originality-Novelty:** 3
**Q2-2 Correctness-Technical Quality:** 3
**Q2-5 Clarity Of Writing:** 3

**Q1 Summary And Contributions:**

The paper proposes a new algorithm for finding solutions to POMDPs with designated goal states and the objective to maximize the probability of reaching them.

**Q2-3 Extent To Which Claims Are Supported By Evidence:**

3: Good: the main claims are supported by convincing evidence (in the form of adequate experimental evaluation, proofs, (pseudo-)code, references, assumptions).

**Q2-4 Reproducibility:**

3: Good: key resources (e.g. proofs, code, data) are available and key details (e.g. proofs, experimental setup) are sufficiently well-described for competent researchers to confidently reproduce the main results.

**Q3 Main Strengths:**

A new algorithm for POMDPs and MRPP is proposed, after an analysis of the reasons of reasons why some state-of-the-art methods for standard discounted-rewards POMDP methods do not fare well for MRPP.

**Q4 Main Weakness:**

The experimental validation is narrow and therefore not very convincing.

**Q5 Detailed Comments To The Authors:**

The "heuristic search" in the title could be replaced with something a bit more informative. Most methods use some heuristics.

Page 1: The paper ascribes undecidability of POMDPs to Papadimitriou and Tsitsiklis (1987), but the only result in that paper about partial observability shows PSPACE-hardness for a limited POMDP problem. The first proof of undecidability of POMDPs appears to be Madani, Hanks and Condon (1999/2003).

You should cite the original work(s) on point-based POMDP methods rather than an overview article.

"The evolution of a POMDP is in fact an infinite MDP": Wouldn't it be more accurate and informative to say that POMDPs can be reduced to MDPs with an infinite number of states?

The temporal logic inspired notation with the diamond is not the most intuitive one for most of the readers.

Do not write "Sec" for "Section". You could also write "Fig." and "Alg." in full, just for clarity.

The pseudocode in Algorithm 1 ignores too many details to make sufficient sense. Please improve this, quite a bit!

The experimentation uses a very small number of different problems, which is unfortunate.

I am not sure the comparison to SARSOP for Q1 is as informative as it could be. The problems being experimented have really small state spaces, and the discount factors for SARSOP are very or extremely close to 1. Wouldn't similar solutions be obtained also a bit lower gammas (e.g. 0.99, 0.98, 0.95), with correspondingly (much?) lower runtimes?

In Table 1, the "number of belief states explored" is only given for STORM and Ours. The use of boldface for "best" results is a bit inconsistent, as e.g. the best runtime is not always the one that is in boldface. What do you exactly mean by "best"?

Bibliography has a lot of issues: Acronyms written in lower case, proper names written with lower-case initials. In BibTeX, you have to surround upper-case letters with braces {POMDP} to guarantee that they are correctly shown in upper case.

Bibliography: Why are there web links, DOIs as well as ISSN and ISBN numbers to some of the papers? Those should be removed.

Bibliography: A number of of entries have very incomplete publication data, including Andriusshchenko et al. 2022, Coquelin and Munos (2007), Alfaro (1998), Smith and Simmons (2005).

Bibliography: "K.J Astrom" has a period missing after "J".

**Q9 Complying With Reviewing Instructions:**

Yes

---

> ### Author Rebuttal · Authors · 2024-04-05
>
> We thank the reviewer for your careful review and constructive feedback.
>
> **Q4: The experimental validation is narrow and therefore not very convincing.**
>
> For comparison with related work, we focused on existing MRPP benchmark problems that induce infinite belief MDP, which are more difficult problems than those that induce finite belief MDPs. Hence, we used all such benchmark problems with the $P_{max}$ query presented in related papers: STORM (Bork et al. 2020), PAYNT (Andriushchenko et al. 2022), and SAYNT (Andriushchenko et al. 2023).
>
> Per reviewers' suggestions, we performed additional evaluations. All the new results can be found in our official comment to all reviewers above. For more details, please refer to our response to Reviewer zRvP (Q4.3/Q5.1).
>
> **Q5: SARSOP: Wouldn't similar solutions be obtained with a bit lower gammas (e.g. 0.99, 0.98, 0.95), with correspondingly (much?) lower runtimes?**
>
> For many POMDP algorithms such as HSVI/SARSOP, the discount factor can be used as a tuning parameter, where discount is increased closer to 1 if more accuracy is required. Therefore, we decided to present a set of experiments that answer Q1 using varying discount factors that are very close to 1. We have run additional experiments to analyze the effects of even lower discounting. The results for $\gamma=$ 0.99, 0.98, 0.95 can be found in the official comments to all reviewers above and the anonymized Github repository: [https://github.com/UAISubmission746/HSVIRP](https://github.com/UAISubmission746/HSVIRP)
>
> From the results, we see that just setting $\gamma$ slightly lower can lead to very poor performance, even though it generally leads to lower runtimes (albeit not always!). This poor performance of SARSOP for MRPP experimentally validates our discussion in Section 3 and our proposed modifications. For MRPP, the optimal policy can be difficult to find using existing trial-based algorithms designed for discounted problems since the sparse (goal) reward may only be obtained after a long horizon, even for problems with small state spaces.
>
> **Q5: The pseudocode in Algorithm 1 ignores too many details to make sufficient sense.**
>
> Thank you for the feedback. Since the page limit allows for 2 extra pages for the final submission, we will add more details in the pseudocode for Algorithm 1 and add a schematic of the algorithm, to make the overall HSVI-RP algorithm and ideas easier to follow.
>
> **Q5: In Table 1, the "number of belief states explored" is only given for STORM and Ours.**
>
> STORM, Overapp, and ours are mainly belief expansion-based, and so the number of belief states explored is most relevant for Q2 and thus reported. The number of beliefs for Overapp in Refuel20 was mistakenly omitted - it should have 177K beliefs. PAYNT does not expand beliefs, and although SAYNT expands beliefs, it does not report the number of beliefs expanded. The number of grid belief points for PRISM is 15 (grid-av 4-0.1), 97 (grid-av 10-0.3), and 6K (Refuel6). These are reflected in the updated tables in the linked repository.
>
> **Q5: What is meant by "best"?**
>
> "Best" is the result that achieves the best value, and lowest runtime if multiple methods achieve the same value. We apologize for a typo in the entry Refuel6, where the time taken for STORM and PAYNT are swapped. STORM should be 1.4s (and bold) and PAYNT should be 77.8s. These results are in line with the results reported in the original papers. We will fix the table in the final version. An updated table can be found at [https://github.com/UAISubmission746/HSVIRP](https://github.com/UAISubmission746/HSVIRP).
>
> **Q5: Suggestions / Citation / Bibliography /  Notation**
>
> Thank you for the suggestions and catching these inaccuracies. We will update the citation of undecidability to (Madani, Hanks and Condon 2003), and cite the original works when introducing point-based methods. We will clarify the reduction of POMDPs to MDPs with infinite number of states. We will also correct the bibliography issues.

---

### Official Review · Reviewer_Y18H · 2024-03-22

**Q2-1 Originality-Novelty:** 2
**Q2-2 Correctness-Technical Quality:** 3
**Q2-5 Clarity Of Writing:** 3

**Q1 Summary And Contributions:**

This paper studies heuristic, trial-based methods for solving POMDPs, targeting the Maximal Reachability Probability Problem (MRPP) objective, i.e. maximizing the probability to reach a set of states. It adapts the well known HSVI algorithm, which is usually applied to discounted reward POMDPs, to MRPP, by making several modifications. A theoretical convergence of the lower bound is given. Experimental results are presented and show that the authors' technique performs well compared to existing tools.

**Q2-3 Extent To Which Claims Are Supported By Evidence:**

2: Fair: the main claims are somewhat supported by evidence (but the experimental evaluation may be weak, or does not match entirely with the claims, important baselines may be missing, proofs contain important ideas but lack rigor, algorithmic details are only discussed superficially, references are imprecise, assumptions are not sufficiently motivated or explicated, etc.).

**Q2-4 Reproducibility:**

2: Fair: key resources (e.g. proofs, code, data) are unavailable but key details (e.g. proof sketches, experimental setup) are sufficiently well-described for an expert to confidently reproduce the main results.

**Q3 Main Strengths:**

+ The problem tackled is well motivated: there are interesting applications, for example, in the formal methods community

+ Based on results so far, it seems this approach has the potential to perform well, and to outperform the various existing tools.

+ Related work in the area is described clearly and puts the current work in context

**Q4 Main Weakness:**

- I feel space is needlessly used on content that doesn't really add to the contribution. In particular,  most of of Sec 3 seems largely known/obvious. First, the reduction of MRPP to expected total reward must, by definition, be undiscounted and it is clear that it is incorrect for discount < 1. So the counterexample/proof for Proposition 1 seems unnecessary. It also seems strange to present this as trial-based methods giving "incorrect bounds". Similarly, the lack of convergence when the discount = 1 is also expected, since lambda < 1 is a requirement for existing results.

- The theoretical contribution feels a bit rushed/unfinished. Theorem 1 is presented briefly but without much clarity/detail. (E.g. the term "action selection radius" was  never introduced. Why does the type of optimal policy matter?) More importantly, saying nothing about the upper bound feels like an important omission and limits the usefulness of the lower bound convergence result. It would also also be good to have a clear statement clear about what differs wrt normal HSVI.

- The first part of the experiments (Q1 in Sec 5) seems unnecessary. We already know that using non-1 discounts here doesn't work, so why use them? And in Table 1, we do not know whether the gap between lower/upper bounds is due to the discount or to SARSOP, so we seem to learn nothing.

- The experimental results are presented for a relatively small set of benchmarks (three). The various tools/papers compared to and cited use a larger set. Why are these not considered?

- No attempt is made to assist reproducibility.

**Q5 Detailed Comments To The Authors:**

- p.3 why is absolute value needed if the optimal value is always greater?

- p.4 what is a "blind policy"? What is V_MDP? What is WEU?

**Q9 Complying With Reviewing Instructions:**

Yes

---

> ### Author Rebuttal · Authors · 2024-04-05
>
> Thank you for your helpful and constructive review. We are pleased you share our perspective on our modifications to HSVI and its potential.
>
> **Q4.1: Section 3 seems largely known/obvious.**
>
> We agree that discussions on discount and reward structure are unnecessary for a reader familiar with MRPP. However, we felt the need to include Section 3 as it provides justification/insight for our modifications to HSVI2 for MRPP. Our second point about trial termination with $\gamma = 1$ directly motivates an adaptively search depth trial termination strategy, while the third point motivates the graph representation, search heuristics, and upper bound backup techniques. While discounting under-approximating the true value is not a surprising result, we argue that a discussion that discounting can lead to arbitrarily bad solutions is valuable. Of note, these points are not considered in (Bouton et al. 2020) when using SARSOP for MRPP; hence our decision to include them for clarity.
>
> **Q4.2: Theorem 1 clarity/detail.**
>
> We will provide a detailed discussion of lower bound convergence. The action selection radius $\xi$ is introduced in Eq. (10). The type of optimal policy matters since it is impossible to approximate an optimal policy which requires infinite memory even as iterations goes to infinity, due to the inapproximability of undiscounted infinite horizon POMDPs (Madini, Hanks, and Condon, AIJ'03). HSVI-RP's lower bound convergence is contingent on being able to search trials of finite depth, as depth increases to infinity.
>
> **Q4.2: Upper bound Analysis.**
>
> Thanks for the feedback. Initially, we didn't include this analysis because we can only guarantee convergence of the upper bound for certain problems, but cannot generalize it. For instance, we can prove that the upper bound converges for the POMDPs that induce finite belief MDPs. However, the upper bound does not converge for MRPPs that lowering the upper bound requires reaching a belief state through an unbounded number of state transitions starting from the initial belief. This is related to the undecidability of POMDPs. Nonetheless, we emphasize that HSVI-RP provides sound upper bounds that guide the search for optimal policies. Note that among the compared algorithms, only PRISM provides guarantees on asymptotic convergence, while the other algorithms only provably provide sound bounds.
>
> Per reviewer's comment, we will add this discussion and more in-depth analysis in the final version.
>
> **Q4.2: Clear statement about what differs wrt normal HSVI.**
>
> That's a good suggestion. The proof of HSVI2 relies heavily on discounting to bound the trial depths required. Loops are also not an issue due discounting. In contrast, the HSVI-RP's convergence for MRPP stems from the graph representation, termination criteria and trial-based expansion technique, which allow adequate exploration of the belief MDP. We will add this discussion in the final version.
>
> **Q4.3: Necessity of Q1 in Section 5.**
>
> For many POMDP algorithms such as HSVI/SARSOP, the discount factor can be used as a tuning parameter, where discount is increased if more accuracy is required. Q1 analyzes the effects of this straightforward method for MRPP, and empirically validates our modifications.
>
> SARSOP was chosen as it is shown to performs better than HSVI. There may be multiple reasons for the resulting bound gaps, but Table 1 mainly shows that this method can perform poorly, even with discount very close to 1. Without these results, it is unclear that current trial-based methods are practically insufficient. Case in Point: Reviewer 3Pxw has further questions about using even lower discount factors to achieve similar performance, and lowering $\gamma$ more yields very poor performance (additional experiments in our official comment above).
>
> **Q4.4: The various tools/papers compared to and cited use a larger set. Why are these not considered?**
>
> For comparison with related work, we focused on existing MRPP benchmark problems that induce infinite belief MDP, which are more difficult problems than those that induce finite belief MDPs. Hence, we used all such benchmark problems with the $P_{max}$ query presented in related papers: STORM, PAYNT, and SAYNT.
>
> Per reviewers' suggestions, we performed additional evaluations. All the new results can be found in our official comment to all reviewers above. For more details, please refer to our response to Reviewer zRvP (Q4.3/Q5.1).
>
> **Q4.5: Reproducibility.**
>
> We have open sourced our code and benchmark results at [https://github.com/UAISubmission746/HSVIRP](https://github.com/UAISubmission746/HSVIRP)
>
> **Q5.1: Why is absolute value needed?**
>
> It is not necessary; We will remove it.
>
> **Q5.2: What is a blind policy, $V_{MDP}$, WEU?**
>
> We apologize for the lack of clarity. WEU is short for Weighted Excess Uncertainty Eq. (3), and blind policy and $V_{MDP}$ are discussed in Section 4. We will make sure to introduce them properly.

---

### Official Review · Reviewer_4Qcq · 2024-03-23

**Q2-1 Originality-Novelty:** 2
**Q2-2 Correctness-Technical Quality:** 3
**Q2-5 Clarity Of Writing:** 4

**Q1 Summary And Contributions:**

The authors consider the problem of computing values for reachability properties over POMDPs, which is an instance of the Maximal Reachability Probability Problem. In order to achieve that, they propose a new algorithm called Heuristic Search Value Iteration for Reachability Problems (HSVI-RP), a variant of Heuristic Search Value Iteration (HSVI). An empirical evaluation is presented where they compare the performance of a prototype implementation against various existing tools.

**Q2-3 Extent To Which Claims Are Supported By Evidence:**

3: Good: the main claims are supported by convincing evidence (in the form of adequate experimental evaluation, proofs, (pseudo-)code, references, assumptions).

**Q2-4 Reproducibility:**

2: Fair: key resources (e.g. proofs, code, data) are unavailable but key details (e.g. proof sketches, experimental setup) are sufficiently well-described for an expert to confidently reproduce the main results.

**Q3 Main Strengths:**

The paper is generally well-written, the problem is clearly defined and quite relevant to the UAI community. The performance results are encouraging, showing improvement in computational time, memory usage and tightness of bounds in most instances of 3 different benchmark problems. Public availability of an implementation of the algorithm be would be beneficial.

**Q4 Main Weakness:**

I do not see a whole lot of novelty here. It is a mix of largely pre-existing techniques put together into a variant of an algorithm that was already known to be effective in similar problems. Crucially, an analysis of upper bound convergence is missing. Furthermore, a performance comparison with Goal-HSVI, which perhaps would be the most relevant, is not included in any capacity (I appreciate it wouldn't be a direct comparison). The fact that it does better than large, general model checking tools such as STORM and PRISM is reassuring but not surprising. Finally, very little insight is given about the impact of the different parameters in the performance of the algorithm and there are virtually no comments on strategy synthesis.

**Q5 Detailed Comments To The Authors:**

Minor typos/ suggested changes:

Page 1 - "...transition and observation uncertainty" -> "..transition and observation uncertainties"

Page 1 - (unbounded-time) MRPP / (indefinite-horizon) MRPP

Page 3 - "Probabilistic reachability probability can..." -> "Probabilistic reachability values can..."

Page 8 - SAYNT, which integrates belief exploration of STORM and FSC generation of PAYNT and leverages the strengths of each approach, and performs the best among the these algorithms. -> Needs rephrasing.

**Q9 Complying With Reviewing Instructions:**

Yes

---

> ### Author Rebuttal · Authors · 2024-04-05
>
> We thank the reviewer for the thoughtful comments and suggestions.
>
> **Q3: Public availability of algorithm.**
>
> We have open sourced (and anonymized) our code and benchmark results at [https://github.com/UAISubmission746/HSVIRP](https://github.com/UAISubmission746/HSVIRP)
>
> **Q4: Novelty.**
>
> We agree that trial-based search for POMDPs is known to be effective. But, we believe that HSVI-RP is a novel advancement in the literature for MRPP for the following reasons: (i) There is no prior work that uses trial-based search for the undiscounted MRPP. The undiscounted and sparse reward nature of MRPP makes many common techniques for POMDPs unsuitable. In Section 3, we discuss specific aspects of why existing trial-based algorithms perform poorly for MRPP, which justifies and motivates our proposed modifications. (ii) existing methods for MRPP are mainly developed by the formal methods community, where probabilistic guarantees ($\gamma = 1$) is of utmost importance. On the other hand, trial-based methods are developed by the AI and robotics communities, where $\gamma < 1$ is typically considered. This work bridges these communities by drawing techniques from both and developing a new algorithm that significantly improves the state of the art.
>
> **Q4: Missing analysis on upper bound convergence.**
>
> Thanks for the feedback. Initially, we didn't include this analysis because we can only guarantee convergence of the upper bound for certain problems, but cannot generalize it. For instance, we can prove that the upper bound converges for the POMDPs that induce finite belief MDPs. However, the upper bound does not converge for MRPPs that lowering the upper bound requires reaching a belief state through an unbounded number of state transitions starting from the initial belief. This is related to the undecidability of POMDPs. Nonetheless, we emphasize that HSVI-RP provides sound upper bounds that guide the search for optimal policies. Note that among the compared algorithms, only PRISM (Norman et al. 2017) provides asymptotic convergence guarantees, while the other algorithms only provably provide sound bounds.
>
> Per reviewer's comment, we'll add this discussion and more in-depth analysis in the final version.
>
> **Q4: Performance comparison with Goal-HSVI.**
>
> Thanks for this comment, giving us an opportunity to expand. We considered comparing against Goal-HSVI since it seemed very relevant to our work. But, after digging into details, we realized that Goal-HSVI cannot be used for MRPP. Goal-HSVI requires that the target state is reachable with probability 1, and the problem is to minimize costs that are strictly positive. These assumptions do not hold for the reward structure induced by MRPP. In fact, goal POMDPs (solved by Goal-HSVI) and an MRPP are both special but distinct cases of the stochastic shortest path problem, but different modifications have to be made to HSVI to work for each problem.
>
> **Q4: HSVI-RP vs large, general model checking tools such as STORM and PRISM.**
>
> We agree that STORM and PRISM are established model checking tools, and encompass more than just POMDP problems. But, we emphasize that they are the state-of-the-art algorithms for MRPP and other undiscounted objectives. The compared algorithms in STORM and PRISM are specialized algorithms for POMDPs with indefinite horizon objectives.
>
> **Q4: Impact of the different parameters.**
>
> Thank you for the comment. We will provide a discussion on parameters in the final version. Below, we provide a summary.
>
> Parameter $\xi$: defines a radius around the best upper bound within which actions are considered. Lower $\xi$ favor actions with higher upper bounds, improving upper bounds faster, but may reduce efficiency by limiting exploration.
>
> Parameters $c_a$ and $c_z$: exploration constants for action and observation selection. Higher values encourage more exploration but can hinder convergence if set too high due to too little exploitation.
>
> Parameter $n$: ratio of exploration trials to Upper Bound Value Iterations (VI). While VI is crucial to improve upper bounds, it has large computation overhead as VI has to be conducted for all nodes in the belief graph. $n$ balances exploration and attempting to improve upper bounds (costly step).
>
> Parameters $d_{trial}$ and $d_{inc}$: control the rate of increase of search depth. Higher values are beneficial for long horizon problems but may slow search efficiency if increased too quickly.
>
> **Q4: On strategy synthesis.**
>
> As discussed in Section 2, the optimal value function can be under-approximated arbitrarily well by a set of alpha vectors. This set of alpha vectors implicitly represent the policy. Since $V^*(b) \geq V^{\pi}(b) = \max_{\alpha \in \Gamma}(\alpha^T b)$, the action at belief $b$ is chosen using $\arg\max_{\alpha \in \Gamma}(\alpha^T b)$.
>
> **Q5: Typos/suggestions**
>
> Thanks for catching the errors and suggesting these changes. We'll update the paper accordingly.

---

### Official Review · Reviewer_zRvP · 2024-03-23

**Q2-1 Originality-Novelty:** 3
**Q2-2 Correctness-Technical Quality:** 3
**Q2-5 Clarity Of Writing:** 2

**Q10 Ethical Concerns:**

None.

**Q1 Summary And Contributions:**

This work proposes a new algorithm for solving the maximal reachability probability problem (MRPP) in POMDP's.  The proposed method provides both improving upper and improving lower bounds allowing for a measure of confidence to be ascertained.  Included are several proofs that demonstrate the inability of forming upper bounds when using discounted rewards with existing trial-based search algorithms and Theorem that the lower bound asymptotically converges to the probability given an optimal policy.

Empirical evaluation is done on several existing benchmarks comparing overall performance (ie. accuracy of the returned probabilities) and bound qualities of several methods.  The proposed method seems to perform best in most cases.

**Q2-3 Extent To Which Claims Are Supported By Evidence:**

2: Fair: the main claims are somewhat supported by evidence (but the experimental evaluation may be weak, or does not match entirely with the claims, important baselines may be missing, proofs contain important ideas but lack rigor, algorithmic details are only discussed superficially, references are imprecise, assumptions are not sufficiently motivated or explicated, etc.).

**Q2-4 Reproducibility:**

2: Fair: key resources (e.g. proofs, code, data) are unavailable but key details (e.g. proof sketches, experimental setup) are sufficiently well-described for an expert to confidently reproduce the main results.

**Q3 Main Strengths:**

* the work addresses a particularly difficult problem in the space of problems for POMDP's
* justification for claims were provided
* compared various aspects of their algorithm to other methods
* compared with several other methods

**Q4 Main Weakness:**

* perhaps the writing could be simplified in some places
* presentation could benefit from additional figures / schematics
* limited number of problems experimented on

**Q5 Detailed Comments To The Authors:**

1. Can you please comment on how the problems were chosen and why they are good representatives?
2. Are there cases where you predict your algorithm would suffer?
3. Can you please comment a bit more about the different algorithms compared against, and how they were parameterized (especially regarding Q3, but also SARSOP Kurnaiwati et al. [2008] with Bouton et al. [2020]).
4. When comparing bounds, do any of the competing algorithms produce anytime bounds?
5. Discounting rewards only produces incorrect upper bounds, right?
6. The regret bound $\epsilon$ for MRPP should be in $(0,1)$ not $(0,\infty)$, right?

* I wonder if the explanations can be simplified in some places and additional figures help supplement descriptions

**Q9 Complying With Reviewing Instructions:**

Yes

---

> ### Author Rebuttal · Authors · 2024-04-05
>
> Thank you for the encouraging comments and insightful questions. We are encouraged that the reviewer finds the work well founded and that our empirical results support the benefits of our proposed method.
>
> **Q4.1-2: Additional figures/schematics**
>
> That's a very good suggestion. Since the page limit allows for 2 extra pages for the final submission, we will utilize that space to add a schematic of the algorithm, together with more details in the pseudocode for Algorithm 1 to make the overall HSVI-RP algorithm and ideas in the paper easier to follow.
>
> **Q4.3/Q5.1: Limited number of problems experimented on. Can you please comment on how the problems were chosen and why they are good representatives?**
>
> To evaluate our algorithm against related work, we focused on existing MRPP benchmark problems that induce infinite belief MDP, which are more difficult problems than those that induce finite belief MDPs.  Hence, we used all such benchmark problems with the $P_{max}$ query presented in related papers: STORM (Bork et al. 2020), PAYNT (Andriushchenko et al. 2022), and SAYNT (Andriushchenko et al. 2023).
>
> Per reviewers' suggestions, we performed additional evaluations on the benchmark problems Crypt4 and Nrp8, introduced in the PRISM paper (Norman et al. 2017), which induce finite belief MDPs. To further enrich the evaluations, we also benchmark our algorithm on the RockSample problem in Bouton et al. 2020 ($\phi_2)$.
>
> All the new results can be found in our official comment to all reviewers above.  These results show that our algorithm (HSVI-RP) generally remains the best amongst existing methods. This is in spite the fact that our method computes both lower and upper bounds together, whereas other methods (except PRISM) compute only one of the bounds. Specifically, on Crypt4, our algorithm takes $<1s$ to reach the optimal lower bound, and another 15.6s for the upper bound to converge. Since Crypt4 has many observations, more iterations are required to effectively search the space.
>
> **Q5.2: Are there cases where you predict your algorithm would suffer?**
>
> As discussed in the evaluation section, iterations of HSVI-RP can take a long time in very large problems which require deep trials. This occurs due to the large constructed belief graph and size of the alpha-vector set. The algorithm is also less effective when there are many long loops, where the upper bounds are not very informative.
>
> Additionally, the trial-based method is less efficient in problems with a large branching factor or when many actions/observations have similar values since many iterations are required to effectively search the space. In the discounted POMDP literature for trial-based search, there are some approaches, such as PLEASE (Zhang et al. 2015), which attempt to combat this issue by creating additional branches during a trial. This could be an interesting future direction.
>
> **Q5.3: Can you please comment a bit more about the different algorithms compared against, and how they were parameterized.**
>
> We have provided a short summary of each algorithm in the related work and experiments sections. PAYNT uses a counter-example guided inductive synthesis approach by searching in the space of finite state controllers using a combination of abstraction-refinement and counterexamples. SAYNT uses reference policies from STORM to accelerate synthesis of PAYNT, and uses the policies obtained from PAYNT to improve belief expansion search for STORM. SARSOP works similarly to HSVI2, with different heuristics during each exploration trial. Unfortunately, due to space constraints, we defer the algorithm details to the cited works. We used the toolboxes provided in the cited papers. These toolboxes provided a set of recommended parameter settings. We reported the best parameters among the available settings for the compared algorithms, available in the data files in the Github repository. The results were similar to those reported in the original papers. For SARSOP, we used the default parameters in the tool, which generally perform well for discounted POMDP problems.
>
> **Q5.4: When comparing bounds, do any of the competing algorithms produce anytime bounds?**
>
> Except PRISM and STORM, all the competing algorithms are able to produce anytime bounds in the sense that their computed bounds monotically improve using an abstraction-refinement procedure. However, since all these methods use some sort of abstraction-refinement procedure, each abstraction iteration may require some time. For the compared algorithms, we reported the lowest runtime to reach the best value.
>
> **Q5.5: Discounting rewards only produces incorrect upper bounds.**
>
> Yes, thank you for catching this inaccuracy. We will update the text.
>
> **Q5.6: The regret bound for MRPP should be (0,1)**
>
> Yes, thank you for noticing this. (0,$\infty$) is redundant as the probability values are bounded between 0 and 1. We will update the text accordingly.

---

### Meta-Review · Area_Chair_tZk8 · 2024-04-15

**Summary:** The paper considers the problem of maximizing the probability of reaching certain target states in POMDPs, a challenging flavor of decision-making problem where rewards are sparse and the discount factor is one. The main challenge in this type of problem typically lies in efficient, and theoretically sound, exploration of the state space (which is not fully observable). The paper proposes, analyzes, and empirically evaluates a novel (value-based search) algorithm, based on Heuristic Search Value Iteration, where the new algorithm performs well.

**Recommendation:** After the discussion phase we have 4 'weak accept' and one 'reject', but the corresponding reviewer has confirmed in the discussion that their criticism was sufficiently addressed to not object acceptance. Putting all the information together, the paper seems technically sound and claims are supported by evidence. The weakpoints of the paper are clarity and presentation (the paper is quite technical after all), and somewhat incremental novelty. Overall I think the work is ready and interesting, but the fact that it only has 'weak accepts' and no reviewer championing the paper is also somewhat symptomatic of the work - it is a borderline case. I am currently leaning slightly towards recommending acceptance if there are not too many more publications with higher scores - out of my batch this paper is the weakest 'accept'. Should it not make the cut, then I think it can definitely be turned into a strong publication with a bit more work and a major revision based on the reviewer feedback.